# How Sparse Can We Prune A Deep Network: A Fundamental Limit Perspective

**Qiaozhe Zhang, Ruijie Zhang, Jun Sun,*  Yingzhuang Liu**

School of EIC, Huazhong University of Science and Technology
`qiaozhezhang@hust.edu.cn, ruijiezhang@ucsb.edu`
`juns@hust.edu.cn, liuyz@hust.edu.cn`

## Abstract

Network pruning is a commonly used measure to alleviate the storage and computational burden of deep neural networks. However, the fundamental limit of network pruning is still lacking. To close the gap, in this work we'll take a first-principles approach, i.e. we'll directly impose the sparsity constraint on the loss function and leverage the framework of *statistical dimension* in convex geometry, thus enabling us to characterize the sharp phase transition point, which can be regarded as the fundamental limit of the pruning ratio. Through this limit, we're able to identify two key factors that determine the pruning ratio limit, namely, *weight magnitude* and *network sharpness*. Generally speaking, the flatter the loss landscape or the smaller the weight magnitude, the smaller pruning ratio. Moreover, we provide efficient countermeasures to address the challenges in the computation of the pruning limit, which mainly involves the accurate spectrum estimation of a large-scale and non-positive Hessian matrix. Moreover, through the lens of the pruning ratio threshold, we can also provide rigorous interpretations on several heuristics in existing pruning algorithms. Extensive experiments are performed which demonstrate that our theoretical pruning ratio threshold coincides very well with the experiments. All codes are available at: `https://github.com/QiaozheZhang/Global-One-shot-Pruning`

## 1 Introduction

Deep neural networks (DNNs) have achieved huge success in the past decade, which relies heavily on the overparametrization, i.e. the number of parameters is normally several order of magnitudes more than the number of data samples. Though being a key enabler for the striking performance of DNN, overparametrization poses huge burden for computation and storage in practice. It is therefore tempting to ask: 1) Can we prune the DNN by a large ratio without performance sacrifice? 2) What's the fundamental limit of network pruning?

For the first question, a key approach is to perform network pruning, which was first introduced by [21]. Network pruning can substantially decrease the number of parameters, thus alleviating the computational and storage burden. The basic idea of network pruning is simple, i.e., to devise metrics to evaluate the significance of parameters and remove the insignificant ones. Various pruning algorithms have been proposed so far: [21; 12; 13; 23; 42; 35; 36; 24; 22; 15] and [13].

In contrast, our understanding on the second question, i.e., the fundamental limit of network pruning, is far less. Some relevant works are: [19] proposed to characterize the degrees of freedom of a DNN by exploiting the framework of Gaussian width. [25] directly applied the above degrees of freedom result to the pruning problem, in the main purpose of unveiling the mechanisms behind the

---

*Jun Sun (juns@hust.edu.cn) is the corresponding author.

38th Conference on Neural Information Processing Systems (NeurIPS 2024).

Lottery Thicket Hypothesis (LTH) [6]. The lower bound of pruning ratio is briefly mentioned in [25], unfortunately, their predicted lower bound does not match the actual value well, in some cases even with big gap. And there is no discussion on the upper bound in that paper.

Despite the above progress, a systematic of the study on the **fundamental limit** of network pruning is still lacking. To close this gap, we'll take a first principles approach to address this problem and exploit the powerful framework of the high-dimensional convex geometry. Essentially, we impose the sparsity constraint directly on the loss function, thus we can reduce the *pruning limit* problem to a *set intersection* problem, then we leverage the powerful approximate kinematics formula [1] in convex geometry, which provides a very sharp phase transition point to easily address the set intersection problem, thus enabling a very tight characterization of the limit of network pruning. Intuitively speaking, the limit of pruning is determined by the dimension of the loss sublevel set (whose definition is in Sec. 2) of the network, the higher the latter, the smaller the former.

The key **contributions** of this paper can be summarized as follows:

- We fully characterize the limit of network pruning, which coincides perfectly with the experiments. Moreover, this limit conveys two valuable messages: 1) The smaller the *network sharpness* (defined as the trace of the Hessian matrix), the more we can prune the network; 2) The smaller the *weight magnitude*, the more we can prune the network.

- We provide an efficient *spectrum estimation* algorithm for *large-scale* Hessian matrices when computing the Gaussian width of a high-dimensional *non-convex* set.

- We present intuitive explanations on many heuristics utilized in existing pruning algorithms through the lens of our pruning ratio limit, which include: (a). Why gradually changing the pruning ratio during iterative pruning is preferred. (b). Why employing $l_2$ regularization makes significant performance difference in Rare Gems algorithm [28]. (c).Why magnitude pruning might be the optimal pruning strategy.

## 1.1 Related Work

**Pruning Methods:** Unstructured pruning involves removing unimportant weights without adhering to some structural constraints. Typical methods in this class include: [13] presented the train-prune-retrain method, which reduces the storage and computation of neural networks by learning only the significant connections. [37; 38] employed the energy consumption of each layer as the metric to determine the pruning order and developed latency tables to identify the layers that should be pruned. [11] proposed dynamic network surgery, which reduced network complexity significantly by pruning connections in real time. [6] proposed pruning by iteratively removing part of the small weights, and based on Frankle's iterative pruning[6], [28] introduced $l_2$-norm to constrain the magnitude of unimportant parameters during iterative training. To the best of our knowledge, there is still no *systematic* study on the fundamental limit of pruning from the theoretical perspective.

**Understanding Neural Networks through Convex Geometry:** Convex Geometry is a powerful tool for characterizing the performance limit of high-dimensional statistical inference [5] and learning problems. For statistical inference, [5] pioneered to employ the convex geometry to study the recovery threshold of the classical linear inverse problem. For statistical learning, [19] studied the training dimension threshold of the network from a geometric point of view by utilizing the Gordon's Escape theorem[10], which shows that the network can be trained with less degrees of freedom (DoF) than the network size in the affine subspace. The most relevant work to ours is [25], which studied the Lottery Tickets Hypothesis (LTH)[6] by applying the above DoF results in [19] to demonstrate that iteration is needed in LTH and that pruning is impacted by the eigenvalues of the loss landscape. The main difference between [25] and ours are as follows: 1) The lower bound of the pruning ratio is only briefly mentioned in [25], and their predicted lower bound does not match the actual value well, in some cases even with quite big gap (the main reason lies in the spectrum estimation error in their algorithm). 2) The core results in [25] are based on Gordon's Escape theorem [10], which can only provide the lower bound (necessary condition). 3) Rigorous analysis as well as the computational issues regarding the pruning limit are lacking in [25]. In contrast, all the above issues are addressed in this paper.

## 2 Problem Setup & Key Notions

To explore the fundamental limit of network pruning, we'll take the first principles approach. In specific, we directly impose the sparsity constraint on the original loss function, thus the feasibility of pruning can be reduced to determining whether two sets, i.e. *the sublevel set* (determined by the Hessian matrix of the loss function) and the *k-sparse set* intersects. Through this framework, we're able to leverage tools in high-dimensional convex geometry, such as statistical dimension [1], Gaussian width [34] and Approximate Kinematics Formula [1].

**Model Setup.** Let $\hat{\mathbf{y}} = f(\mathbf{w}, \mathbf{x})$ be a deep neural network with weights $\mathbf{w} \in \mathbb{R}^D$ and inputs $\mathbf{x} \in \mathbb{R}^K$. For a given training data set $\{\mathbf{x}_n, \mathbf{y}_n\}_{n=1}^N$ and a loss function $\ell$, the empirical loss landscape is defined as $\mathcal{L}(\mathbf{w}) = \frac{1}{N} \sum_{n=1}^N \ell(f(\mathbf{w}, \mathbf{x}_n), \mathbf{y}_n)$.

**Pruning Objective.** In essence, network pruning can be formulated as the following optimization problem:

$$\min \|\mathbf{w}\|_0 \quad \text{s.t.} \quad \mathcal{L}(\mathbf{w}) \leq \mathcal{L}(\mathbf{w}^*) + \epsilon \tag{1}$$

where $\mathbf{w}$ is the pruned weight and $\mathbf{w}^*$ is the original one.

**Sparse Network.** Given a dense network with weights $\mathbf{w}^*$, we denote its sparse counterpart as a $k$-sparse network, whose weight is given by: $\mathbf{w}^k = \mathbf{w}^* \odot \mathbf{m}$, where $\odot$ is element-wise multiplication and $\|\mathbf{m}\|_0 = k$.

**Loss Sublevel Sets.** A loss sublevel set of a network is the set of all weights $\mathbf{w}$ that achieve the loss up to $\mathcal{L}(\mathbf{w}^*) + \epsilon$:

$$S(\epsilon) := \{\mathbf{w} \in \mathbb{R}^D : \mathcal{L}(\mathbf{w}) \leq \mathcal{L}(\mathbf{w}^*) + \epsilon\}. \tag{2}$$

**Feasible $k$-Sparse Pruning.** We define the **pruning ratio** as $\rho = k/D$ and call a sparse weight $\mathbf{w}^k$ as a feasible $k$-sparse pruning if it satisfies:

$$S(\epsilon) \cap \{\mathbf{w}^k\} \neq \emptyset, \tag{3}$$

Below are some key notions and results from high dimensional convex geometry, which are of critical importance to our work.

**Definition 2.1 (Convex Cone & Conic Hull)** *A convex cone $\mathcal{C} \in \mathbb{R}^D$ is a convex set that satisfy: $\sum_i \eta_i x_i \in \mathcal{C}$ for all $\eta_i > 0$ and $x_i \in \mathcal{C}$. The convex conic hull of a set $S$ is defined as:*

$$\mathcal{C}(S) := \{\sum_i \eta_i \mathbf{w}_i \in \mathbb{R}^D : \eta_i > 0, \ \mathbf{w}_i \in S\} \tag{4}$$

**Definition 2.2 (Gaussian Width [34])** *The gaussian width of a subset $S \in \mathbb{R}^D$ is given by:*

$$w(S) = \frac{1}{2} \mathbb{E} \sup_{\mathbf{x}, \mathbf{y} \in S} \langle \mathbf{g}, \mathbf{x} - \mathbf{y} \rangle, \mathbf{g} \sim \mathcal{N}(\mathbf{0}, \mathbf{I}_{D \times D}). \tag{5}$$

Gaussian width is useful to characterize the complexity of a convex body. On the other hand, statistical dimension is an important metric to characterize the complexity of convex cones. Intuitively speaking, the bigger the cone, the larger the statistical dimension, as illustrated in Fig. 1(b).

**Definition 2.3 (Statistical Dimension [1])** *The statistical dimension $\delta(\mathcal{C})$ of a convex cone $\mathcal{C}$ is:*

$$\delta(\mathcal{C}) := \mathbb{E}[\|\Pi_{\mathcal{C}}(\mathbf{g})\|_2^2] \tag{6}$$

*where $\Pi_{\mathcal{C}}$ is the Euclidean metric projector onto $\mathcal{C}$ and $\mathbf{g} \sim \mathcal{N}(\mathbf{0}, \mathbf{I}_{D \times D})$ is a standard normal vector.*

To characterize the sufficient and necessary condition of the set intersection, we'll resort to the powerful Approximate Kinematics Formula [1], which basically says that for two convex cones (or generally, sets), if the sum of their statistical dimension exceeds the ambient dimension, these two cones would intersect with probability 1, otherwise they would intersect with probability 0.

**Theorem 2.4 (Approximate Kinematics Formula, Theorem 7.1 of [1])** *Let $\mathcal{C}$ be a convex conic hull of a sublevel set $S(\epsilon)$ in $\mathbb{R}^D$, and draw a random orthogonal basis $\mathbf{Q} \in \mathbb{R}^{D \times D}$. For a $k$-dimensional subspace $S_k$, it holds that:*

$$\delta(\mathcal{C}) + k \lesssim D \Rightarrow \mathbb{P}\{\mathcal{C} \cap \mathbf{Q}S_k = \emptyset\} \approx 1$$
$$\delta(\mathcal{C}) + k \gtrsim D \Rightarrow \mathbb{P}\{\mathcal{C} \cap \mathbf{Q}S_k = \emptyset\} \approx 0$$

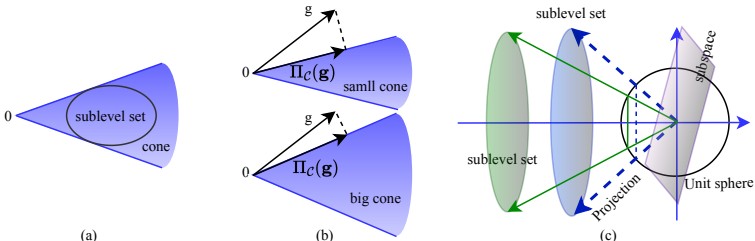

Figure 1: **Panel (a, b):** Illustration of a convex conic hull and the statistical dimension. **Panel (c):** Effect of projection distance on projection size and intersection probability.

## 3 Bounds of Pruning Ratio

### 3.1 Lower Bound of Pruning Ratio

In this section, we aim to characterize the lower bound of pruning ratio, i.e. when the pruning ratio falls below some threshold, it's *impossible* to retain the generalization performance. To establish this impossibility result, we'll leverage the Approximate Kinematics Formula as detailed in Theorem 2.4.

#### 3.1.1 Network Pruning As Set Intersection

To demonstrate that when $k$ is smaller than a given threshold, it is impossible to find a performance-preserving $k$-sparse network induced by the dense network, we need to prove that the loss sublevel set has no intersection with any $k$-sparse set resulting from the dense weight, i.e. $S(\epsilon) \cap \{\mathbf{w}^k\} = \emptyset$.

To that end, it suffices to prove its translated version, namely $S_{\mathbf{w}^k}(\epsilon) \cap \{\mathbf{0}\} = \emptyset$, where $S_{\mathbf{w}^k}(\epsilon) := \{\mathbf{w} - \mathbf{w}^k : \mathbf{w} \in S(\epsilon)\}$. To prove the latter, we'll further prove its strengthened version, i.e. the convex conic hull of $S_{\mathbf{w}^k}(\epsilon)$ and a random orthogonal rotation of the subspace $S(\mathbf{w}^k)$, which is comprised of all vectors that share the same zero-pattern as $\mathbf{w}^k$ (including the point $\mathbf{0}$), has no intersection. Namely, we'll prove that

$$\mathcal{C}(S_{\mathbf{w}^k}(\epsilon)) \cap \mathbf{Q}S(\mathbf{w}^k) = \emptyset, \tag{7}$$

where $\mathbf{Q}$ denotes the Haar-measured orthogonal rotation.

To prove Eq.7, we can easily invoke the necessary condition of the Approximate Kinematics Formula (Theorem 2.4). In order to calculate the involved statistical dimension therein, we choose to calculate the corresponding Gaussian width as the proxy by taking advantage of the following theorem.

**Theorem 3.1 (Gaussian Width vs. Statistical Dimension, Proposition 10.2 of [1])** *Given a unit sphere* $\mathbb{S}^{D-1} := \{\mathbf{x} \in \mathbb{R}^D : \|\mathbf{x}\| = 1\}$*, let* $\mathcal{C}$ *be a convex cone in* $\mathbb{R}^D$*, then:*

$$w(\mathcal{C} \cap \mathbb{S}^{D-1})^2 \leq \delta(\mathcal{C}) \leq w(\mathcal{C} \cap \mathbb{S}^{D-1})^2 + 1 \tag{8}$$

To calculate the Gaussian width of $\mathcal{C}(S_{\mathbf{w}^k}(\epsilon))$, we need to project the sublevel set $S_{\mathbf{w}^k}(\epsilon)$ onto the surface of the unit sphere centered at origin. which is defined as

$$\mathrm{p}(S_{\mathbf{w}^k}(\epsilon)) = \{(\mathbf{x} - \mathbf{w}^k)/\|\mathbf{x} - \mathbf{w}^k\|_2 : \mathbf{x} \in S_{\mathbf{w}^k}(\epsilon)\}, \tag{9}$$

and illustrated in Fig. 1(c). It is easy to see that as the distance $\|\mathbf{x} - \mathbf{w}^k\|_2$ increases, the projected Gaussian width will decrease, as a result the statistical dimension of the set will also decrease, thus increasing the difficulty of its intersecting with a given subspace.

**Theorem 3.2 (Lower Bound of Pruning Ratio)** *Let* $\mathcal{C}$ *be a convex conic hull of a sublevel set* $S_{\mathbf{w}^k}(\epsilon)$ *in* $\mathbb{R}^D$*.* $\mathbf{w}^k$ *doesn't constitute a feasible $k$-sparse pruning with probability 1, if the following holds:*

$$w(p(S_{\mathbf{w}^k}(\epsilon)))^2 + k \lesssim D \tag{10}$$

This theorem tells us that when the dimension $k$ of the sub-network is lower than $k_L = D - w(\mathrm{p}(S_{\mathbf{w}^k}(\epsilon)))^2$, the subspace will not intersect with $S_{\mathbf{w}^k}(\epsilon)$, i.e., no feasible $k$-sparse pruning can be

found. Therefore, the lower bound of the pruning ratio of the network can be expressed as:

$$\rho_L = \frac{D - w(\text{p}(S_{\mathbf{w}^k}(\epsilon)))^2}{D} = 1 - \frac{w(\text{p}(S_{\mathbf{w}^k}(\epsilon)))^2}{D}. \tag{11}$$

It's worth mentioning that this lower bound has been provided in [25] by utilizing the Gordon's Escape Theorem[10]. The main difference between our work and [10] lies in that Gordon's Escape Theorem is not strong enough to provide the upper bound (sufficient condition) of the pruning ratio, while the Approximate Kinematic Formula we employ does.

**Reformulation of the Sublevel Set.** Consider a well-trained deep neural network model with weights $\mathbf{w}^*$ and a loss function $\mathcal{L}(\mathbf{w})$, where $\mathbf{w}$ lies in a small neighborhood of $\mathbf{w}^*$. By performing a Taylor expansion of $\mathcal{L}(\mathbf{w})$ at $\mathbf{w}^*$, using the fact that the first derivative is equal to $\mathbf{0}$ and ignoring the higher order terms, the loss sublevel set $S(\epsilon)$ can be reformulated as:

$$S(\epsilon) = \{\hat{\mathbf{w}} \in \mathbb{R}^D : \frac{1}{2}\hat{\mathbf{w}}^T \mathbf{H} \hat{\mathbf{w}} \leq \epsilon\} \tag{12}$$

where $\hat{\mathbf{w}} = \mathbf{w} - \mathbf{w}^*$ and $\mathbf{H}$ denote the Hessian matrix of $\mathcal{L}(\mathbf{w})$ w.r.t. $\mathbf{w}$. Due to the positive-definiteness of $\mathbf{H}$, $S(\epsilon)$ corresponds to an ellipsoid. The related proofs can be found in Appendix D.1.

### 3.1.2 Gaussain Width of the Ellipsoid

We leverage tools in high-dimensional probability, especially the concentration of measure, which enables us to present a rather precise expression for the Gaussian width of a high-dimensional ellipsoid.

**Lemma 3.3** *For an ellipsoid $S(\epsilon)$ defined by : $S(\epsilon) := \{\mathbf{w} \in \mathbb{R}^D : \frac{1}{2}\mathbf{w}^T \mathbf{H} \mathbf{w} \leq \epsilon\}$, where $\mathbf{H} \in \mathbb{R}^{D \times D}$ is a positive definite matrix, its Gaussian width is given by:*

$$w(S(\epsilon)) \approx (2\epsilon \text{Tr}(\mathbf{H}^{-1}))^{1/2} = (\sum\nolimits_{i=1}^{D} r_i^2)^{1/2} \tag{13}$$

*where $r_i = \sqrt{2\epsilon/\lambda_i}$ is the radius of ellipsoidal body and $\lambda_i$ is the $i$-th eigenvalue of $\mathbf{H}$.*

The proof of Lemma 3.3 is in Appendix D.1. The Gaussian width of an ellipsoid has been provided in [19] as in the interval $[(\sqrt{\frac{2}{\pi}}(\sum_{i=1}^{D} r_i^2)^{1/2}, (\sum_{i=1}^{D} r_i^2)^{1/2}]$, in contrast we sharpen the estimation of Gaussian width to a point $(\sum_{i=1}^{D} r_i^2)^{1/2}$. For the settings which involve projection, the squared radius $r_i^2$ should be modified to $\frac{r_i^2}{\|\mathbf{w}^* - \mathbf{w}^k\|_2^2 + r_i^2}$ [19]. Therefore, the Gaussian width of projected $S(\epsilon)$ defined in Eq.(12) equals:

$$w(\text{p}(S_{\mathbf{w}^k}(\epsilon))) \approx \left(\sum_{i=1}^{D} \frac{r_i^2}{\|\mathbf{w}^* - \mathbf{w}^k\|_2^2 + r_i^2}\right)^{1/2} \tag{14}$$

### 3.1.3 Computable Lower Bound of Pruning Ratio

Combining Eq.(11) and Eq.(14), we obtain the following computable lower bound of the pruning ratio:

**Corollary 3.4** *Given a well-trained deep neural network with trained weight $\mathbf{w}^* \in \mathbb{R}^D$ and a loss function $\mathcal{L}(\mathbf{w})$, for a $k$-sparse pruned weight $\mathbf{w}^k$, the lower bound of pruning ratio of model is:*

$$\rho_L = 1 - \frac{1}{D} \sum_{i=1}^{D} \frac{r_i^2}{\|\mathbf{w}^* - \mathbf{w}^k\|_2^2 + r_i^2}. \tag{15}$$

*where $r_i = \sqrt{2\epsilon/\lambda_i}$ and $\lambda_i$ is the eigenvalue of the Hessian matrix of $\mathcal{L}(\mathbf{w})$ w.r.t. $\mathbf{w}$.*

### 3.1.4 Pruning Ratio vs Magnitude & Sharpness

It is evident from Eq.(15) that for a given trained network (whose spectrum of the Hessian matrix is fixed), to minimize the lower bound of the pruning ratio, we just need to minimize $\|\mathbf{w}^* - \mathbf{w}^k\|_2$, i.e. the sum of magnitudes of the pruned parameters. Therefore, the commonly-used magnitude-based pruning algorithms get justified. Moreover, it also inspires us to employ the one-shot magnitude pruning algorithm as detailed in Section 4, whose performance proves to be better than other existing algorithms, to the best of our knowledge.

Besides the above-discussed magnitude of the pruned sub-vector, we also identify another important factor that determines the pruning ratio, i.e. the *network sharpness*, which describes the sharpness of the loss landscape around the minima, as defined by the trace of the Hessian matrix, namely $\mathrm{Tr}(\mathbf{H})$ ([26] and [9], network flatness is the opposite of network sharpness; as sharpness increases, flatness decreases, and vice versa.).

**Lemma 3.5 (Pruning Ratio vs. Sharpness)** *Given a well-trained neural network $f(\mathbf{w})$, where $\mathbf{w}$ is the parameters. The lower bound of the pruning ratio and the sharpness obeys:*

$$\rho_L \leq 1 - \frac{2\epsilon D}{\|\mathbf{w}^* - \mathbf{w}^k\|_2^2 \mathrm{Tr}(\mathbf{H}) + 2\epsilon D} \tag{16}$$

*where $\mathbf{H}$ is the hessian matrix of the loss function w.r.t. $\mathbf{w}$.*

Lemma 3.5 is obtained by utilizing the Cauchy–Schwarz Inequality, whose proof can be found in Appendix F. It can be seen from Lemma 3.5 that the lower bound of the network pruning ratio is heavily dependent on the sharpness of the network, i.e. flatter networks imply more sparsity. This can be a valuable guideline for both training and pruning the networks. Intuitively, a flatter loss landscape is less sensitive to weight perturbations, indicating greater tolerance to weight removal.

### 3.2 Upper Bound of Pruning Ratio

In order to establish the upper bound of the pruning ratio, we need to prove that there *exists* an $k$-sparse weight vector intersects with the loss sub-level set.

For a given trained weight $\mathbf{w}^*$, we split it into two parts, i.e. the unpruned subvector, $\mathbf{w}^1 = [\mathbf{w}_1^*, \mathbf{w}_2^*, \ldots, \mathbf{w}_k^*]$ and the pruned one $\mathbf{w}^2 = [\mathbf{w}_{k+1}^*, \mathbf{w}_{k+2}^*, \ldots, \mathbf{w}_D^*]$. By fixing $\mathbf{w}^1$, the loss sublevel set can be reformulated as:

$$S(\mathbf{w}^{'}, \epsilon) = \{\mathbf{w}^{'} \in \mathbb{R}^{D-k} : \mathcal{L}([\mathbf{w}^1, \mathbf{w}^{'}]) \leq \mathcal{L}(\mathbf{w}^*) + \epsilon\} \tag{17}$$

In order to prove the existence of a $k$-sparse weight vector $\mathbf{w}^k$, we just need to show that the all-zero vector is in $S(\mathbf{w}^{'}, \epsilon)$. To that end, we'll take advantage of the sufficient condition of the approximate kinematics formula (Theorem 2.4) to show that it suffices to render the statistical dimension of the projected cone of $S(\mathbf{w}^{'}, \epsilon)$ being full, i.e. $D - k$. Thus we can obtain the upper bound of the number of unpruned parameters, i.e. $k$.

Specifically, by invoking the sufficient part of Theorem 2.4, the upper bound of the pruning ratio by a given pruning strategy is as follows:

**Theorem 3.6 (Upper Bound of Pruning Ratio)** *Given a sublevel set $S(\mathbf{w}^{'}, \epsilon)$ in $\mathbb{R}^{D-k}$. To ensure that the all-zero vector $\mathbf{0} \in \mathbb{R}^{D-k}$ contained in $S(\mathbf{w}^{'}, \epsilon)$, it suffices that:*

$$w(p(S(\mathbf{w}^{'}, \epsilon)))^2 \gtrsim D - k.$$

The Gaussian width of projected $S(\mathbf{w}^{'}, \epsilon)$ can be easily obtained by employing Lemma 3.3, i.e. $w(p(S(\mathbf{w}^{'}, \epsilon)))^2 = \sum_i^{D-k} \frac{\widetilde{r}_i^2}{\|\mathbf{w}^* - \mathbf{w}^k\|_2^2 + \widetilde{r}_i^2}$, where $\widetilde{r}_i = \sqrt{2\epsilon/\widetilde{\lambda}_i}$, $\widetilde{\lambda}_i$ is the eigenvalue of the hessian matrix of $\mathcal{L}([\mathbf{w}^1, \mathbf{w}^{'}])$ w.r.t. to $\mathbf{w}^{'}$ and the fact that $\|\mathbf{w}^* - \mathbf{w}^k\|_2 = \|\mathbf{w}^2\|_2$ is used. Correspondingly, the upper bound of the pruning ratio can be expressed as

$$\rho_U \approx 1 - \frac{w(p(S(\mathbf{w}^{'}, \epsilon)))^2}{D} = 1 - \frac{1}{D} \sum_{i=1}^{D-k} \frac{\widetilde{r}_i^2}{\|\mathbf{w}^* - \mathbf{w}^k\|_2^2 + \widetilde{r}_i^2}. \tag{18}$$

### 3.3 Fundamental Limit of Pruning Ratios

As demonstrated above, the pruning ratio can be bounded as follows:

$$1 - \frac{1}{D} \sum_{i=1}^{D} \frac{r_i^2}{\|\mathbf{w}^* - \mathbf{w}^k\|_2^2 + r_i^2} \leq \rho \leq 1 - \frac{1}{D} \sum_{i}^{D-k} \frac{\widetilde{r}_i^2}{\|\mathbf{w}^* - \mathbf{w}^k\|_2^2 + \widetilde{r}_i^2}. \tag{19}$$

It is easy to notice that the upper bound and lower bound are of nearly identical form. In fact, as we'll elaborate in the following, they are also of quite close value, which implies that we are able to obtain a sharp characterization of the fundamental limit of pruning ratio. Meanwhile, it is worthwhile noting that the pruning limit depends on the magnitude of the final weights, which might be significantly impacted by the weight initialization. Therefore, we still need to explore whether the magnitude of final weights is dependent on the initialization values. In the appendix, we'll demonstrate that once the data, network architecture and training method are fixed, the distribution of trained network weights remains nearly insensitive to the initializations, thus yielding an affirmative answer about the above question.

## 4 Achievable Scheme & Computational Issues

Thus far we have established the lower bound and upper bound of the pruning ratio by leveraging the Approximate Kinematic Formula in convex geometry [1]. To proceed, we will demonstrate that our obtained bounds are tight in the sense that we can devise an achievable pruning algorithm whose corresponding upper bound is quite close to the lowest possible value of the lower bound. As argued in Corollary 3.4, the magnitude pruning, which removes all the smallest $D - k$ weights, will result in the lowest pruning ratio lower bound. Inspired by this result, we'll focus on the magnitude pruning methods in order to approach the lower bound in the sequel.

For the lower bound part, we need to address several challenges regarding the computation of the Gaussian width of a high-dimensional deformed ellipsoid, which involves tackling the *non-positiveness* of the Hessian matrix as well as the *spectrum estimation* of a large-scale Hessian matrix.

For the upper bound part, we'll focus on a *relaxed* version of Eq. 1 by introducing the $l_1$ regularization term, for the sake of computational complexity. Then we'll employ the *one-shot* magnitude pruning to compress the network.

### 4.1 Computational Challenges & Countermeasures

To compute the lower bound of the pruning ratio, we need to address the following two challenges:

**Gaussian Width of the Deformed Ellipsoid.** In practice, it is usually hard for the network to converge to the exact minima, thus leading to a non-positive definite Hessian matrix. In other words, the ideal ellipsoid gets deformed due to the existence of negative eigenvalues. Determining the Gaussian width of the deformed ellipsoid is a challenging task. To address this problem, we resort to convexifying (i.e. taking the convex hull of) the deformed ellipsoid and then calculating the Gaussian width of the latter instead, by proving that the convexifying procedure has no impact on the Gaussian width. (The proof is presented in Appendix D.2).

**Improved Spectrum Estimation.** Neural networks often exhibit a quite significant number of zero-valued or vanishingly small eigenvalues in their Hessian matrices. It's hard for the spectrum estimation algorithm SLQ (Stochastic Lanczos Quadrature) proposed by [39] to obtain accurate estimation of these small eigenvalues. To address this issue, we propose to enhance the existing large-scale spectrum estimation algorithms by a key modification, i,e, to estimate the number of these exceptionally small eigenvalues by employing the Hessian matrix sampling. See Algorithm 1 for the details of the improved spectrum Estimation algorithm. A comprehensive description of the algorithm and its experimental results are presented in Appendix C.

### 4.2 Achievable Scheme: $l_1$ Regularization & One-shot Magnitude Pruning

Inspired by the lower bound as well as upper bound of the pruning ratio, in which the magnitude of pruning parameters plays a key role, it's sensible to focus on the magnitude-based pruning methods.

**Algorithm 1** Improved SLQ (Stochastic Lanczos Quadrature) Spectrum Estimation Algorithm

---

**Input:** Hermitian matrix $\mathbf{A}$ of size $n \times n$, Lanczos iterations $m$, ESD computation iterations $l$, gaussian kernel $f$ and variance $\sigma^2$, sampling num $s$ and row $l_1$ norm threshold $\epsilon$.
**Output:** The spectral distribution of matrix $\mathbf{A}$
**for** $i = 2, ..., l$ **do**
    1). Get the tridiagonal matrix $\mathbf{T}$ of size $m \times m$ through Lanczos algorithm[18];
    2). Compute $m$ eigenpairs $(\lambda_k^{(i)}, v_k^{(i)})$ from $\mathbf{T}$;
    3). $\phi_\sigma^i(t) = \sum_{k=1}^m \tau_k^{(i)} f(\lambda_k^{(i)}; t, \sigma)$, where $\tau_k^{(i)} = (v_k^{(i)}[1])^2$ is the first component of $v_k^{(i)}$;
**end for**
4). Random sample $s$ rows $\mathbf{A}_i$ of matrix $\mathbf{A}$ and calculate every $l_1$ norm $\|\mathbf{A}_i\|_1$ by take $i \in [1, s]$.
5). Compute the number $z$ of $\|\mathbf{A}_i\|_1$ which satisfy $\|\mathbf{A}_i\|_1 \leq \epsilon$.
6). Compute the integration of $\frac{1}{l} \sum_{i=1}^l \phi_\sigma^i(t)$, termed as $c$, $\phi(t) = \frac{1}{l} \sum_{i=1}^l \phi_\sigma^i(t) + \frac{cz}{s-z} \delta(1^{-30})$.
**Return:** $\phi(t)$

---

On the other hand, to find exact solutions of our original problem for the best pruning in Eq. 1, it is obviously very hard due to the existence of $l_0$ norm. To make it feasible, it's natural to perform a convex relaxation of $l_0$ norm, namely, by employing $l_1$ regularization instead. Aside from the computational advantage of this relaxation, it is worthy noting that $l_1$ regularization provides two extra benefits: 1) A large portion of eigenvalues of the trained Hessian matrix are zero-valued or of quite small value, which renders the calculation of the pruning limit more accurately and fast. 2) A large portion of trained weights are of quite small value, thus making the lower bound and upper bound very close. Detailed statistics about the eigenvalues and magnitudes can be found in Figure 6.

Specifically, by utilizing the Lagrange formulation and convex relaxation of $l_0$ norm, the pruning objective in Eq.1 can be reformulated as:

$$\min \ \mathcal{L}(\mathbf{w}) + \lambda \|\mathbf{w}\|_1 \tag{20}$$

After training with this relaxed objective, the network weights will be pruned based on magnitudes *one time*, rather than in an iterative way as in [13; 6; 28]. The performance of the above described pruning scheme (termed as "$l_1$ *regularized one-shot magnitude pruning*" and abbreviated as "LOMP") can be found in Table 10 in Appendix. The above stated "zero-dominating" property due to $l_1$ regularization gets supported in Figure 2(b), where it can be seen that the majority of weights are indeed extremely small.

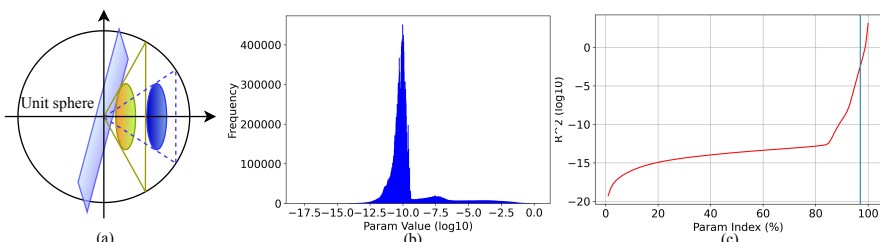

(a)               (b)               (c)

Figure 2: Effect of extremely small projection distance on projection size and intersection probability and statistics of ResNet50 on TinyImagenet. Statistics regarding all experiments can be found in Appendix G.

The above "zero-domination" property turns out to be of critical value for our proposed pruning scheme to nearly achieve the limit (lower bound) of the pruning ratio. Fig. 2(c) illustrates the curve $\|\mathbf{w}^2\|_2^2$, i.e. the $l_2$ norm of the $D - k$ smallest weights, w.r.t. $k/D$. The vertical line therein represents $\rho_L$, the lower bound of the pruning ratio predicted in Section 3.1. When $k = D\rho_L$, the curve and the line will intersect as shown in Figure 2(c). Mathematically, the upper bound for the pruning ratio can be approximated as follows:

$$\rho_U = 1 - \frac{1}{D} \sum_{i=1}^{D-k} \frac{\widetilde{r}_i^2}{\|\mathbf{w}^2\|_2^2 + \widetilde{r}_i^2} \approx 1 - \frac{1}{D} \sum_{i=1}^{D-k} \frac{\widetilde{r}_i^2}{\widetilde{r}_i^2} = \frac{k}{D} = \rho_L \tag{21}$$

It can be seen from the above demonstration that the upper bound corresponding to our proposed pruning scheme almost *coincides* with the minimal lower bound! In other words, we have established the fundamental limit of the pruning ratio. To provide further validation of the above claim, we performed the experiments five times across eight tasks and reported the differences between the upper bound and lower bound, denoted as $\Delta$, in Table 9.

Table 1: The Difference Between Lower Bound and Upper Bound of Pruning Ratio.

| CIFAR10 | FC5 | FC12 | Alexnet | VGG16 |
|---|---|---|---|---|
| $\Delta(\%)$ | 0.17±0.05 | 0.05±0.03 | 0.02±0.01 | 0.01±0.00 |
| **ResNet** | **18 on CIFAR100** | **50 on CIFAR100** | **18 on TinyImagenet** | **50 on TinyImagenet** |
| $\Delta(\%)$ | 0.12±0.05 | 0.11±0.09 | 0.09±0.01 | 0.27±0.22 |

# 5 Experiments

In this section, we validate our pruning method as well as the theoretical limit of the pruning ratio by experiments.

**Tasks.** We evaluate the pruning ratio threshold on: Full-Connect-5(FC5), Full-Connect-12(FC12), AlexNet [17] and VGG16 [27] on CIFAR10 [16], ResNet18 and ResNet50 [14] on CIFAR100 and TinyImageNet [20]. We employ the $l_1$ regularization during training, and execute a one-shot magnitude-based pruning and assess its performance with various sparsity ratios, in terms of the metrics of accuracy and loss. Detailed descriptions of datasets, networks, hyper-parameters, and eigenspectrum adjustment can be found in Section B of the Appendix. Moreover, the performance comparison between $l_1$-regularization-based magnitude pruning and other pruning methods can be found in Table 10 in Appendix.

## 5.1 Validation of Pruning Lower Bound

After training with the $l_1$ regularization, we compute the eigenvalues and present the theoretical limit of pruning ratio. By pruning the trained network to various sparsity levels, we depict in Figure 3 both the line of theoretical lower bound and the sparsity-accuracy curve for the above-listed tasks. From the figures we can see that our theoretical result matches the numerical pruning ratio quite well.

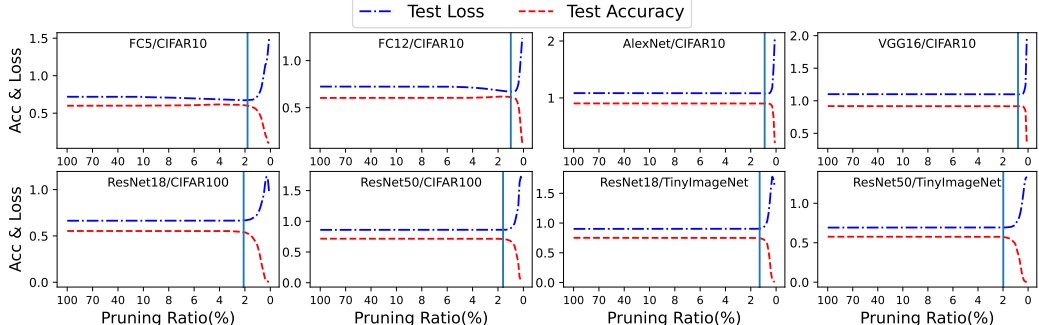

Figure 3: The impact of sparsity on loss and test accuracy are obtained on the test dataset, and we mark the theoretical pruning ratio limit with vertical lines. The loss values have been normalized and translated.

## 5.2 Prediction Performance

We present a more detailed comparison between our theoretical limit of pruning ratio, and the actual values by experiments in Table 2, which shows nearly perfect agreement between them. The difference between the theoretical value and the actual value is donated as $\Delta$.

# 6 Interpretation of Pruning Heuristics

Equipped with the fundamental limit of network pruning, we're now able to provide rigorous interpretations of several heuristics employed by existing pruning algorithms.

Table 2: Comparison between Theoretical and Actual Values of Pruning Ratio

| Dataset | Model | Theo. Value(%) | Actual Value(%) | $\Delta$(%) |
|---|---|---|---|---|
| **CIFAR10** | FC5 | 2.1±0.25 | 1.7±0.12 | -0.40±0.35 |
| | FC12 | 1.0±0.30 | 0.8±0.06 | -0.24±0.33 |
| | AlexNet | 0.9±0.00 | 0.8±0.08 | -0.14±0.08 |
| | VGG16 | 0.8±0.06 | 0.8±0.08 | 0.04±0.08 |
| **CIFAR100** | ResNet18 | 1.5±0.05 | 2.0±0.13 | 0.54±0.15 |
| | ResNet50 | 1.9±0.05 | 2.1±0.16 | 0.28±0.19 |
| **TinyImagenet** | ResNet18 | 3.9±0.82 | 4.3±0.38 | 0.46±0.71 |
| | ResNet50 | 2.6±0.24 | 2.9±0.33 | 0.36±0.10 |

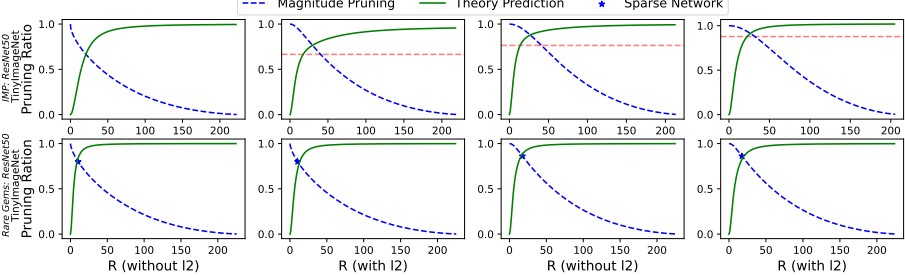

Figure 4: **Top Row:** From left to right, as the number of iterations increases, it leads to an increase in the theoretical pruning ratio threshold. The horizontal line represents the last pruning ratio. **Bottom Row:** The comparison of the pruning ratio threshold between using and not using $l_2$-regularization. Sparse networks are obtained by magnitude-based pruning with fixed pruning ratios. The two plots on the left and the two plots on the right correspond to different fixed pruning ratios. Here, $R = \|\mathbf{w}^* - \mathbf{w}^k\|_2$, which is the projection distance.

**Pruning ratio adjustment is needed in IMP.** For the IMP (Iterative Magnitude Pruning) algorithm [7], we determine the pruning ratio thresholds for various stages through calculations, as depicted in the top row of Figure 4. It is noteworthy that as the pruning depth gradually increases, the theoretical pruning ratio threshold also increases. Therefore, it is appropriate to prune smaller proportions of weights gradually during iterative pruning, Both [43] and [28] have employed pruning rate adjustment, which gradually prunes smaller proportions of the weights with the iteration of the algorithm.

$l_2$**-regularization enhances the performance of Rare Gems.** For the Rare Gems algorithm [28], it is shown that $l_2$ regularization makes a significant difference in terms of the final performance, as shown in the bottom row of Figure 4. The main reason behind this phenomenon is: when $l_2$-regularization is applied, the pruning ratio tends to be larger than the theoretical limit, however, the absence of $l_2$-regularization would result in excessive pruning, which can be regarded as wrong pruning.

## 7 Conclusion

In this paper we explore the fundamental limit of pruning ratio of deep networks by taking the first principles approach and leveraging the framework of convex geometry. Specifically, we reduce the pruning limit problem to the sets intersection problem, and by taking advantage of the powerful Approximate Kinematic Formula, we are able to sharply characterize the fundamental limit of the network pruning ratio. This fundamental limit conveys a key message as follows: the network pruning limit is mainly determined by the *weight magnitude* and the *network sharpness*. These two guidelines can provide intuitive explanations of several heuristics in existing pruning algorithms. Moreover, to address the challenges in computing the involved Gaussian width, we develop an improved spectrum estimation for large-scale and non-positive Hessian matrices. Experiments demonstrate the almost perfect agreement between our theoretical results and the experimental ones.

**Limitations.** In this paper, the (almost) coincidence of the upper bound and lower bound of the pruning ratio depends on the condition that the removed weights are of quite small value, which is enabled by the $l_1$ regularization we employed. Therefore, it is important to demonstrate whether the $l_1$ regularized training is optimal or nearly optimal in the sense of obtaining the smallest sub-network without performance degradation for the original learning problem.

## Acknowledgments and Disclosure of Funding

This work was supported in part by the National Key Research and Development Program of China under Grant 2024YFE0103800, in part by the National Natural Science Foundation of China under Grant 62071192. We thank the anonymous reviewers for their valuable and constructive feedback that helped in shaping the final manuscript.

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

# A  Organization of Appendix

The appendix is organized as follows:

- Sec. A: an overview of the organization of the appendix.
- Sec. B: detail descriptions of the datasets, models, hyper-parameter choices used in our experiments. Additionally, figures illustrating the theoretical pruning threshold are included.
- Sec. C: this section delves into the practical calculation of the Gaussian Width. It addresses the challenges associated with the *non-positiveness* of the Hessian matrix and the *spectrum estimation* of a large-scale Hessian matrix. Experimental results highlighting the "important eigenvalues" are also showcased.
- Sec. D: a comprehensive proof of the Gaussian Width for both the ellipsoid and the deformed ellipsoid is provided.
- Sec. E: this section presents the proof of the "sub-sublevel set" utilized in deriving the upper bound of the pruning ratio. Furthermore, a straightforward explanation of the relationship between the general lower bound and the upper bound is offered.
- Sec. F: omitted proof of the connection between the sharpness and the lower bound of the pruning ratio is detailed in this section.
- Sec. G: performance comparison between the $l_1$ regularized one-shot magnitude pruning, termed as "LOMP", and several prominent pruning strategies. Results from a hypothetical experiment verifying the importance of magnitude in pruning and comprehensive statistical results from Sec. 4.2 are also included.
- Sec. H: limitations of our assumptions and theoretical results.
- Sec. I: broader impacts statement of this research.

# B  Experimental Details

In this section, we describe the datasets, models, hyper-parameter choices and eigenspectrum adjustment used in our experiments. All of our experiments are run using PyTorch 1.12.1 on Nvidia RTX3090s with ubuntu20.04-cuda11.3.1-cudnn8 docker.

## B.1  Dataset

**CIFAR-10.**  CIFAR-10 consists of 60,000 color images, with each image belonging to one of ten different classes with size $32 \times 32$. The classes include common objects such as airplanes, automobiles, birds, cats, deer, dogs, frogs, horses, ships, and trucks. The CIFAR-10 dataset is divided into two subsets: a training set and a test set. The training set contains 50,000 images, while the test set contains 10,000 images [16]. For data processing, we follow the standard augmentation: normalize channel-wise, randomly horizontally flip, and random cropping.

**CIFAR-100.**  The CIFAR-100 dataset consists of 60,000 color images, with each image belonging to one of 100 different fine-grained classes [16]. These classes are organized into 20 superclasses, each containing 5 fine-grained classes. Similar to CIFAR-10, the CIFAR-100 dataset is split into a training set and a test set. The training set contains 50,000 images, and the test set contains 10,000 images. Each image is of size 32x32 pixels and is labeled with its corresponding fine-grained class. Augmentation includes normalize channel-wise, randomly horizontally flip, and random cropping.

**TinyImageNet.**  TinyImageNet comprises 100,000 images distributed across 200 classes, with each class consisting of 500 images [20]. These images have been resized to $64 \times 64$ pixels and are in full color. Each class encompasses 500 training images, 50 validation images, and 50 test images. Data augmentation techniques encompass normalization, random rotation, and random flipping. The dataset includes distinct train, validation, and test sets for experimentation.

## B.2  Model

In all experiments, pruning skips bias and batchnorm, which have little effect on the sparsity of the network. Non-affine batchnorm is applied in the network, and the initialization of the network is kaiming normal initialization.

**Full Connect Network(FC-5, FC-12).**  We train a five-layer fully connected network (FC-5) and a twelve-layer fully connected network FC-12 on CIFAR-10, the network architecture details can be found in Table 3.

Table 3: FC-5 and FC-12 architecture used in our experiments.

| Model | Layer Width |
|---|---|
| FC-5 | 1000, 600, 300, 100, 10 |
| FC-12 | 1000, 900, 800, 750, 700, 650, 600, 500, 400, 200, 100, 10 |

**AlexNet [17].**  We use the standard AlexNet architecture. In order to use CIFAR-10 to train AlexNet, we upsample each picture of CIFAR-10 to $3 \times 224 \times 224$. The detailed network architecture parameters are shown in Table 4.

**VGG-16 [27].**  In the original VGG-16 network, there are 13 convolution layers and 3 FC layers (including the last linear classification layer). We follow the VGG-16 architectures used in [7; 8] to remove the first two FC layers while keeping the last linear classification layer. This finally leads to a 14-layer architecture, but we still call it VGG-16 as it is modified from the original VGG-16 architectural design. Detailed architecture is shown in Table 5. VGG-16 is trained on CIFAR-10.

**ResNet-18 and ResNet-50 [14].**  We use the standard ResNet architecture for TinyImageNet and tune it for the CIFAR-100 dataset. The detailed network architecture parameters are shown in Table 6. ResNet-18 and ResNet-50 is trained on CIFAR-100 and TinyImageNet.

Table 4: AlexNet architecture used in our experiments.

| Layer | Shape | Stride | Padding |
|---|---|---|---|
| conv1 | $3 \times 96 \times 11 \times 11$ | 4 | 1 |
| max pooling | kernel size:3 | 2 | N/A |
| conv2 | $96 \times 256 \times 5 \times 5$ | 1 | 2 |
| max pooling | kernel size:3 | 2 | N/A |
| conv3 | $256 \times 384 \times 3 \times 3$ | 1 | 1 |
| conv4 | $384 \times 384 \times 3 \times 3$ | 1 | 1 |
| conv4 | $384 \times 256 \times 3 \times 3$ | 1 | 1 |
| max pooling | kernel size:3 | 2 | N/A |
| linear1 | $6400 \times 4096$ | N/A | N/A |
| linear1 | $4096 \times 4096$ | N/A | N/A |
| linear1 | $4096 \times 10$ | N/A | N/A |

Table 5: VGG-16 architecture used in our experiments.

| Layer | Shape | Stride | Padding |
|---|---|---|---|
| conv1 | $3 \times 64 \times 3 \times 3$ | 1 | 1 |
| conv2 | $64 \times 64 \times 3 \times 3$ | 1 | 1 |
| max pooling | kernel size:2 | 2 | N/A |
| conv3 | $64 \times 128 \times 3 \times 3$ | 1 | 1 |
| conv4 | $128 \times 128 \times 3 \times 3$ | 1 | 1 |
| max pooling | kernel size:2 | 2 | N/A |
| conv5 | $128 \times 256 \times 3 \times 3$ | 1 | 1 |
| conv6 | $256 \times 256 \times 3 \times 3$ | 1 | 1 |
| conv7 | $256 \times 256 \times 3 \times 3$ | 1 | 1 |
| max pooling | kernel size:2 | 2 | N/A |
| conv8 | $256 \times 512 \times 3 \times 3$ | 1 | 1 |
| conv9 | $512 \times 512 \times 3 \times 3$ | 1 | 1 |
| conv10 | $512 \times 512 \times 3 \times 3$ | 1 | 1 |
| max pooling | kernel size:2 | 2 | N/A |
| conv11 | $512 \times 512 \times 3 \times 3$ | 1 | 1 |
| conv12 | $512 \times 512 \times 3 \times 3$ | 1 | 1 |
| conv13 | $512 \times 512 \times 3 \times 3$ | 1 | 1 |
| max pooling | kernel size:2 | 2 | N/A |
| avg pooling | kernel size:1 | 1 | N/A |
| linear1 | $512 \times 10$ | N/A | N/A |

Table 6: ResNet architecture used in our experiments.

| Layer | ResNet-18 | ResNet-50 |
|---|---|---|
| conv1 | $64, 3 \times 3$; stride:1; padding:1 | $64, 3 \times 3$; stride:1; padding:1 |
| block1 | $\begin{pmatrix} 64, 3 \times 3; \text{stride:1; padding:1} \\ 64, 3 \times 3; \text{stride:1; padding:1} \end{pmatrix} \times 2$ | $\begin{pmatrix} 64, 1 \times 1; \text{stride:1; padding:0} \\ 64, 3 \times 3; \text{stride:1; padding:1} \\ 256, 1 \times 1; \text{stride:1; padding:0} \end{pmatrix} \times 3$ |
| block1 | $\begin{pmatrix} 128, 3 \times 3; \text{stride:2; padding:1} \\ 128, 3 \times 3; \text{stride:1; padding:1} \end{pmatrix} \times 2$ | $\begin{pmatrix} 128, 1 \times 1; \text{stride:1; padding:0} \\ 128, 3 \times 3; \text{stride:2; padding:1} \\ 512, 3 \times 3; \text{stride:1; padding:0} \end{pmatrix} \times 4$ |
| block1 | $\begin{pmatrix} 128, 3 \times 3; \text{stride:2; padding:1} \\ 256, 3 \times 3; \text{stride:1; padding:1} \end{pmatrix} \times 2$ | $\begin{pmatrix} 256, 1 \times 1; \text{stride:1; padding:0} \\ 256, 3 \times 3; \text{stride:2; padding:1} \\ 1024, 1 \times 1; \text{stride:1; padding:0} \end{pmatrix} \times 6$ |
| block1 | $\begin{pmatrix} 512, 3 \times 3; \text{stride:2; padding:1} \\ 512, 3 \times 3; \text{stride:1; padding:0} \end{pmatrix} \times 2$ | $\begin{pmatrix} 512, 1 \times 1; \text{stride:1; padding:1} \\ 512, 3 \times 3; \text{stride:2; padding:1} \\ 2048, 1 \times 1; \text{stride:1; padding:0} \end{pmatrix} \times 3$ |
| avg pooling | kernel size:1; stride:1 | kernel size:1; stride:1 |
| linear1 | $512 \times ClassNum$ | $2048 \times ClassNum$ |

## B.3  Training Hyper-parameters Setup

In this section, we will describe in detail the training hyper-parameters of the Global One-shot Pruning algorithm on multiple datasets and models. The various hyperparameters are detailed in Table 7.

Table 7: Hyper Parameters used for different Datasets and Models.

| Model | Dataset | Batch Size | Epochs | Optimizer | LR | Momentum | Warm Up | Weight Decay | CosineLR | Lambda |
|---|---|---|---|---|---|---|---|---|---|---|
| FC5 | CIFAR10 | 128 | 200 | SGD | 0.01 | 0.9 | 0 | 0 | N/A | 0.00005 |
| FC12 | CIFAR10 | 128 | 200 | SGD | 0.01 | 0.9 | 0 | 0 | N/A | 0.00005 |
| VGG16 | CIFAR10 | 128 | 200 | SGD | 0.01 | 0.9 | 5 | 0 | True | 0.00015 |
| AlexNet | CIFAR10 | 128 | 200 | SGD | 0.01 | 0.9 | 5 | 0 | True | 0.00003 |
| ResNet18 | CIFAR100 | 128 | 200 | SGD | 0.1 | 0.9 | 5 | 0 | True | 0.000055 |
| ResNet50 | CIFAR100 | 128 | 200 | SGD | 0.1 | 0.9 | 5 | 0 | True | 0.00002 |
| ResNet18 | TinyImageNet | 128 | 200 | SGD | 0.01 | 0.9 | 5 | 0 | True | 0.00023 |
| ResNet50 | TinyImageNet | 128 | 200 | SGD | 0.01 | 0.9 | 5 | 0 | True | 0.0001 |

## B.4  Sublevel Set Parameters Setup.

Given a dense well-trained neural network with weighted donated as $\mathbf{w}^*$, the loss sublevel set is defined as $\{\hat{\mathbf{w}} \in \mathbb{R}^D : \frac{1}{2}\hat{\mathbf{w}}^T \mathbf{H}\hat{\mathbf{w}} \leq \epsilon\}$ where $\hat{\mathbf{w}} = \mathbf{w} - \mathbf{w}^*$, under the assumption that the test data is often unavailable and we also generally assume that the training and test data share the same distribution, thus we use the training data to define the loss sublevel set. We compute the standard deviation of the network's loss across multiple batches on the training data set and denote it by $\epsilon$.

Table 8: Hyper Parameters used in SLQ Algorithm.

| Model | Dataset | Runs | Iterations | Bins | Squared Sigma |
|---|---|---|---|---|---|
| FC5 | CIFAR10 | 1 | 128 | 100000 | 1e-10 |
| FC12 | CIFAR10 | 1 | 128 | 100000 | 1e-10 |
| VGG16 | CIFAR10 | 1 | 128 | 100000 | 1e-07 |
| AlexNet | CIFAR10 | 1 | 96 | 100000 | 1e-07 |
| ResNet18 | CIFAR100 | 1 | 128 | 100000 | 1e-07 |
| ResNet50 | CIFAR100 | 1 | 128 | 100000 | 1e-07 |
| ResNet18 | TinyImageNet | 1 | 128 | 100000 | 1e-07 |
| ResNet50 | TinyImageNet | 1 | 88 | 100000 | 1e-07 |

## B.5  Theoretical Pruning Ratio

Taking $\mathbf{w}^*$ as the initial pruning point and calculating the corresponding value of $R = \|\mathbf{w}^k - \mathbf{w}^*\|_2$ for different pruning ratios $k/D$. We then plot the corresponding curve of the theoretically predicted pruning ratio and the calculated $R$ in the same graph. The intersection point of these two curves is taken as the lower bound of the theoretically predicted pruning ratio. All results are shown in Figure 5.

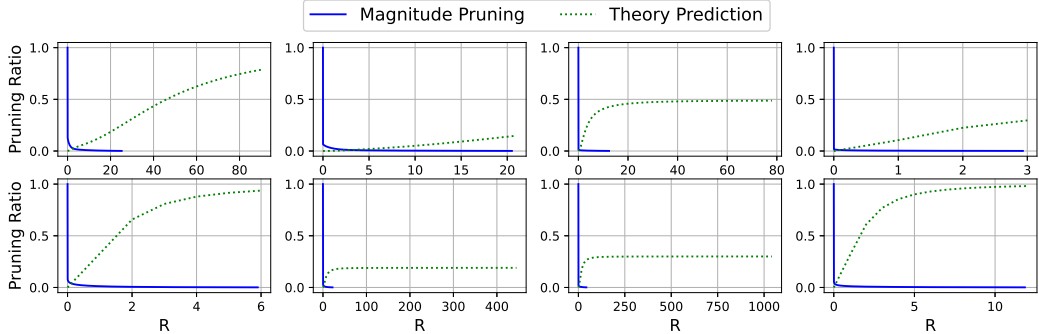

Figure 5: The theoretically predicted pruning ratio in eight tasks. The first row, from left to right, corresponds to FC5, FC12, AlexNet, and VGG16 on CIFAR10. The second row, from left to right, corresponds to ResNet18 and ResNet50 on CIFAR100, as well as ResNet18 and ResNet50 on TinyImagenet.

## C Practical Calculation of Gaussian Width

In practical experiments, determining the Gaussian width of the ellipsoid defined by the network loss function is a challenging task. There are two primary challenges encountered in this section: 1.) the computation of eigenvalues for high-dimensional matrices poses significant difficulty; 2.) the network fails to converge perfectly to the extremum, leading to a non-positive definite Hessian matrix for the loss function. In this section, we tackle these challenges through the utilization of a fast eigenspectrum estimation algorithm and an algorithm that approximates the Gaussian width of a deformed ellipsoid body. These approaches effectively address the aforementioned problems.

### C.1 Improved SLQ (Stochastic Lanczos Quadrature) Spectrum Estimation

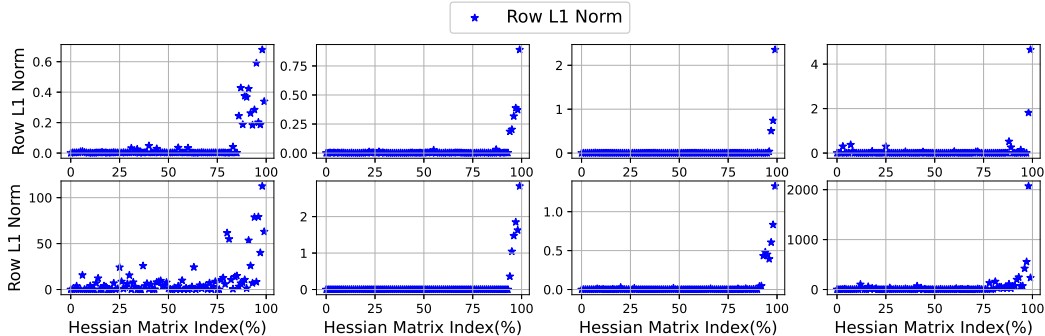

Figure 6: The statistical analysis of the L1 norm of the Hessian matrix in eight tasks. The first row, from left to right, corresponds to FC5, FC12, AlexNet, and VGG16. The second row, from left to right, corresponds to ResNet18 and ResNet50 on CIFAR100, as well as ResNet18 and ResNet50 on TinyImagenet.

Calculating the eigenvalues of large matrices has long been a challenging problem in numerical analysis. One widely used method for efficiently computing these eigenvalues is the Lanczos algorithm[18], which is presented in Algorithm 2. However, due to the huge amount of parameters of the deep neural network, it is still impractical to use this method to calculate the eigenspectrum of the Hessian matrix of a deep neural network. To tackle this problem, [39] proposed SLQ (Stochastic Lanczos Quadrature) Spectrum Estimation Algorithm, which estimates the overall eigenspectrum distribution based on a small number of eigenvalues obtained by Lanczos algorithm. This method enables the efficient computation of the full eigenvalues of large matrices. Algorithm 2 outlines the step-by-step procedure for the classic Lanczos algorithm, providing a comprehensive guide for its implementation. The algorithm requires the selection of the number of iterations, denoted as $m$, which determines the size of the resulting triangular matrix $\mathbf{T}$.

In general, the Lanczos algorithm is not capable of accurately computing zero eigenvalues, and this limitation becomes more pronounced when the SLQ algorithm has a small number of iterations. Similarly, vanishingly small eigenvalues are also ignored by Lanczos. However, in a well-trained large-scale deep neural network, the experiment found that the network loss function hessian matrix has a large number of zero eigenvalues and vanishingly small eigenvalues. In the Gaussian width of the ellipsoid, the zero eigenvalues and vanishingly small eigenvalues have the same effect on the width (insensitive to other parameters), and we collectively refer to these eigenvalues as the "important" eigenvalues. We divide the weight into 100 parts from small to large, calculate the second-order derivative (including partial derivative) of smallest weight in each part, and sum the absolute values of all second-order derivatives of the weight, which corresponds to $l_1$-norm of a row in hessian matrix, and the row $l_1$-norm is zero or a vanishingly small corresponds to an "important" eigenvalue, the experimental results can be seen in the Figure 6, from which the number of missing eigenvalues of the SLQ algorithm can be estimated, we then add the same number of 1e-30 as the missing eigenvalues in the Hessian matrix eigenspectrum. All the SLQ algorithm parameters are discribed in Table 8 and the statistical analysis of the $l_1$ norm of Hessian matrix rows for all experiments is presented in Figure 6. For details of the SLQ algorithm and the improved SLQ algorithm, see Algorithm 3 and Algorithm 1

---

**Algorithm 2** The Lanczos Algorithm

---

**Input:** a Hermitian matrix $\mathbf{A}$ of size $n \times n$, a number of iterations $m$
**Output:** a tridiagonal real symmetric matrix $\mathbf{T}$ of size $m \times m$
initialization:
1. Draw a random vector $\mathbf{v_1}$ of size $n \times 1$ from $\mathcal{N}(0,1)$ and normalize it;
2. $\mathbf{w_1'} = \mathbf{A}\mathbf{v_1}$; $\alpha_1 = <\mathbf{w_1'}, \mathbf{v_1}>$; $\mathbf{w_1} = \mathbf{w_1'} - \alpha_1\mathbf{v_1}$;
3.
**for** $j = 2, ..., m$ **do**
   1). $\beta_j = \|\mathbf{w_{j-1}}\|$;
   2).
   **if** $\beta_j = 0$ **then**
     stop
   **else**
     $\mathbf{v_j} = \mathbf{w_{j-1}}/\beta_j$
   **end if**
   3). $\mathbf{w_j'} = \mathbf{A}\mathbf{v_j}$;
   4). $\alpha_j = <\mathbf{w_j'}, \mathbf{v_j}>$;
   5). $\mathbf{w_j} = \mathbf{w_j'} - \alpha_j\mathbf{v_j} - \beta_j\mathbf{v_{j-1}}$;
**end for**
4. $\mathbf{T}(i,i) = \alpha_i$, $i = 1, \ldots, m$;
   $\mathbf{T}(i,i+1) = \mathbf{T}(i+1,i) = \beta_i$, $i = 1, \ldots, m-1$.
**Return: T**

---

---

**Algorithm 3** SLQ(Stochastic Lanczos Quadrature) Spectrum Estimation Algorithm

---

**Input:** A hermitian matrix $\mathbf{A}$ of size $n \times n$, Lanczos iterations $m$, ESD computation iterations $l$, gaussian kernel $f$ and variance $\sigma^2$.
**Output:** The spectral distribution of matrix $\mathbf{A}$
**for** $i = 2, ..., l$ **do**
   1). Get the tridiagonal matrix $\mathbf{T}$ of size $m \times m$ through Lanczos algorithm;
   2). Compute $m$ eigenpairs $(\lambda_k^{(i)}, v_k^{(i)})$ from $\mathbf{T}$;
   3). $\phi_\sigma^i(t) = \sum_{k=1}^m \tau_k^{(i)} f(\lambda_k^{(i)}; t, \sigma)$, where $\tau_k^{(i)} = (v_k^{(i)}[1])^2$ is the first component of $v_k^{(i)}$.
**end for**
4). $\phi(t) = \frac{1}{l}\sum_{i=1}^l \phi_\sigma^i(t)$
**Return:** $\phi(t)$

---

### C.2 Gaussian Width of the Deformed Ellipsoid

After effective training, it is generally assumed that a deep neural network will converge to the global minimum of its loss function. However, in practice, even after meticulous tuning, the network tends to oscillate around the minimum instead of converging to it. This leads to that the Hessian matrix of the loss function would be non-positive definite, and the resulting geometric body defined by this matrix would change from an expected ellipsoid to a hyperboloid, which is unfortunately nonconvex. To quantify the Gaussian width of the ellipsoid corresponding to the perfect minima, we propose to approximate it by convexifying the deformed ellipsoid through replacing the associated negative eigenvalues with its absolute value. This processing turns out to be very effective, as demonstrated by the experimental results.

**Lemma C.1** *Consider a well-trained neural network with weights* $\mathbf{w}$*, whose loss function defined by* $\mathbf{w}$ *has a Hessian matrix* $\mathbf{H}$*. Due to the non-positive definiteness of* $\mathbf{H}$*, there exist negative eigenvalues. Let the eigenvalue decomposition of* $\mathbf{H}$ *be* $\mathbf{H} = \mathbf{v}^T\mathbf{\Sigma}\mathbf{v}$*, where* $\mathbf{\Sigma}$ *is a diagonal matrix of eigenvalues. Let* $\mathbf{D} = \mathbf{v}^T|\mathbf{\Sigma}|\mathbf{v}$*, where* $|\cdot|$ *means absolute operation. the geometric objects defined by H and D are* $S(\epsilon) := \{\mathbf{w} \in \mathbb{R}^D : \frac{1}{2}\mathbf{w}^T\mathbf{H}\mathbf{w} \leq \epsilon\}$ *and* $\hat{S}(\epsilon) := \{\mathbf{w} \in \mathbb{R}^D : \frac{1}{2}\mathbf{w}^T\mathbf{D}\mathbf{w} \leq \epsilon\}$*, then:*

$$w(S(\epsilon)) \approx w(\hat{S}(\epsilon)) \tag{22}$$

The proof of Lemma C.1 is in Appendix D.2. Lemma C.1 indicates that if the deep neural network converges to a vicinity of the global minimum of the loss function, the Gaussian width of the deformed ellipsoid body can be approximated by taking the convex hull of $S(\epsilon)$.

# D Gaussian Width of the Sublevel Set

In this section, we provide detailed proofs regarding the Gaussian Width of the sublevel sets of quadratic wells.

## D.1 Gaussian Width of the Quadratic Well

Gaussian width is an extremely useful tool to measure the complexity of a convex body. In our proof, we will use the following expression for its definition:

$$w(S) = \frac{1}{2} \mathbb{E} \sup_{\mathbf{x}, \mathbf{y} \in S} \langle \mathbf{g}, \mathbf{x} - \mathbf{y} \rangle, \mathbf{g} \sim \mathcal{N}(\mathbf{0}, \mathbf{I}_{D \times D})$$

Concentration of measure is a universal phenomenon in high-dimensional probability. Basically, it says that a random variable which depends in a smooth way on many independent random variables (but not too much on any of them) is essentially *constant*.[30; 31; 32]

**Theorem D.1 (Gaussian concentration)** *Consider a random vector* $\mathbf{x} \sim \mathcal{N}(\mathbf{0}, \mathbf{I}_{n \times n})$ *and an L-Lipschitz function* $f : \mathbb{R}^n \to \mathbb{R}$ *(with respect to the Euclidean metric). Then for* $t \geq 0$

$$\mathbb{P}(|f(\mathbf{x}) - \mathbb{E}f(\mathbf{x})| \geq t) \leq \epsilon, \quad \epsilon = e^{-\frac{t^2}{2L^2}}.$$

Therefore, if $\epsilon$ is small, $f(\mathbf{x})$ can be approximated as $f(\mathbf{x}) \approx \mathbb{E}f(\mathbf{x}) + \sqrt{-2L^2 \ln \epsilon}$.

**Lemma D.2** *Given a random vector* $\mathbf{x} \sim \mathcal{N}(\mathbf{0}, \mathbf{I}_{n \times n})$ *and the inverse of a positive definite Hessian matrix* $\mathbf{Q} = \mathbf{H}^{-1}$*, where* $\mathbf{H} \in \mathbb{R}^{n \times n}$*, we have:*

$$\mathbb{E}\sqrt{\mathbf{x}^\mathbf{T}\mathbf{Q}\mathbf{x}} \approx \sqrt{\mathbb{E}\mathbf{x}^\mathbf{T}\mathbf{Q}\mathbf{x}}$$

*Proof.*
1.) Concentration of $\mathbf{x}^\mathbf{T}\mathbf{Q}\mathbf{x}$

Define $f(\mathbf{x}) = \mathbf{x}^\mathbf{T}\mathbf{Q}\mathbf{x}$, we have:

$$
\begin{aligned}
f(\mathbf{x}) &= \mathbf{x}^\mathbf{T}\mathbf{Q}\mathbf{x} \\
&= \mathbf{x}^\mathbf{T}\mathbf{U}\mathbf{\Sigma}\mathbf{U}^\mathbf{T}\mathbf{x} && \text{Eigenvalue Decomposition of } \mathbf{Q} : \mathbf{Q} = \mathbf{U}\mathbf{\Sigma}\mathbf{U}^\mathbf{T}. \\
&= \sum_{i=1}^{n} \lambda_i x_i^2 \quad \text{w. p. almost 1.} && \text{Invariance of Gaussian under rotation.}
\end{aligned}
$$

where $\lambda_i$ is the eigenvalue of $\mathbf{Q}$. The lipschitz constant $L_f$ of function $f(\mathbf{x})$ is :

$$L_f = \max(|\frac{\partial f}{\partial \mathbf{x}}|) = \max(|2\lambda_i x_i|)$$

Let $g(x_i) = 2\lambda_i x_i$, whose lipschitz constant is $L_g = |2\lambda_i|$. Invoking Theorem D.1, we have:

$$
\begin{aligned}
g(x_i) &\approx \mathbb{E}g(x_i) + \sqrt{-2(2\lambda_i)^2 \ln \epsilon_1} \\
&= \sqrt{-8\lambda_i^2 \ln \epsilon_1}.
\end{aligned}
$$

Therefore, the lipschitz constant of $f(\mathbf{x})$ can be approximated by:

$$L_f = max(\sqrt{-8\lambda_i^2 \ln \epsilon_1}) = \sqrt{-8\ln \epsilon_1} \lambda_{max}$$

Invoking Theorem D.1 again, we establish the concentration of $f(\mathbf{x})$ as follows:

$$
\begin{aligned}
f(\mathbf{x}) &\approx \mathbb{E}f(\mathbf{x}) + \sqrt{-2(L_f)^2 \ln \epsilon_2} && \text{Theorem } D.1. \\
&= \mathbb{E}f(\mathbf{x}) + 4\sqrt{\ln \epsilon_1 \ln \epsilon_2} \lambda_{max}
\end{aligned}
$$

2.) Jensen ratio of $\sqrt{\mathbf{x^T Q x}}$:

$$\mathbb{E}\sqrt{f(\mathbf{x})} \approx \mathbb{E}\sqrt{\mathbb{E}f(\mathbf{x}) + 4\sqrt{\ln\epsilon_1 \ln\epsilon_2}\lambda_{max}} \qquad \text{Concentration of } f(\mathbf{x}).$$

$$\approx \sqrt{\mathbb{E}f(\mathbf{x})} + \frac{2\sqrt{\ln\epsilon_1 \ln\epsilon_2}\lambda_{max}}{\sqrt{\mathbb{E}f(\mathbf{x})}} \qquad \text{Taylor Expansion.}$$

Therefore, the Jensen ratio of $\sqrt{f(\mathbf{x})}$ can be approximated by:

$$\frac{\mathbb{E}\sqrt{f(\mathbf{x})}}{\sqrt{\mathbb{E}f(\mathbf{x})}} \approx 1 + 2\sqrt{\ln\epsilon_1 \ln\epsilon_2}\frac{\lambda_{max}}{\sum_{i=1}^n \lambda_i} \qquad \text{Hutchinson's method [2; 3]}$$

$$= 1 + \delta$$

If $\mathbf{Q}$ is a Wishart matrix, i.e. $\mathbf{Q} = \mathbf{A}^T\mathbf{A}$, where $\mathbf{A}$ is a random matrix whose elements are independently and identically distributed with unit variance, according to the Marchenko-Pastur law [33], the maximum eigenvalue of $\mathbf{Q}$ is approximately $4n$ and the trace of $\mathbf{Q}$ is approximately $n^2$. Therefore, the above Jensen ratio approaches to 1 with decaying rate $\mathcal{O}(\frac{1}{n})$.

For the inverse of a positive definite Hessian matrix which is of our concern, we take $\epsilon_1 = \epsilon_2 = 10^{-4}$, numerical simulations show that when the dimension $n = 10^5$, the corresponding $\delta$ in the above Jensen ratio is on the order of $10^{-3}$, which is in good agreement with the theoretical value and is arguably negligible. Similar as the case of the above-discussed Wishart matrix, when the dimension $n$ increases, the value of $\delta$ will further decrease.

Consequently, we can conclude that $\mathbb{E}\sqrt{f(\mathbf{x})} \approx \sqrt{\mathbb{E}f(\mathbf{x})}$, i.e. $\mathbb{E}\sqrt{\mathbf{x^T Q x}} \approx \sqrt{\mathbb{E}\mathbf{x^T Q x}}$.

**Definition D.3 (Definition of ball)** *A (closed) ball $B(c,r)$ (in $\mathbb{R}^D$) centered at $c \in \mathbb{R}^D$ with radius $r$ is the set*

$$B(c,r) := \{\mathbf{x} \in \mathbb{R}^D : \mathbf{x}^T\mathbf{x} \le r^2\}$$

*The set $B(0,1)$ is called the unit ball. An ellipsoid is just an affine transformation of a ball.*

**Lemma D.4 (Definition of ellipsoid)** . *An ellipsoid $S$ centered at the origin is the image $L(B(0,1))$ of the unit ball under an invertible linear transformation $L : \mathbb{R}^D \to \mathbb{R}^D$. An ellipsoid centered at a general point $c \in \mathbb{R}^D$ is just the translate $c + S$ of some ellipsoid $S$ centered at the origin.*

*Proof.*

$$L(B(0,1)) = \{\mathbf{Lx} : \mathbf{x} \in B(0,1)\}$$
$$= \{\mathbf{y} : \mathbf{L}^{-1}\mathbf{y} \in B(0,1)\}$$
$$= \{\mathbf{y} : (\mathbf{L}^{-1}\mathbf{y})^T\mathbf{L}^{-1}\mathbf{y} \le 1\}$$
$$= \{\mathbf{y} : \mathbf{y}^T(\mathbf{LL}^T)^{-1}\mathbf{y} \le 1\}$$
$$= \{\mathbf{y} : \mathbf{y}^T\mathbf{Q}^{-1}\mathbf{y} \le 1\}$$

where $\mathbf{Q} = \mathbf{LL}^T$ is **positive definite**.
The radius $r_i$ along principal axis $\mathbf{e}_i$ obeys $r_i^2 = \frac{1}{\lambda_i}$, where $\lambda_i$ is the eigenvalue of $\mathbf{Q}^{-1}$ and $\mathbf{e}_i$ is eigen vector.

**Lemma D.5 (Gaussian width of ellipsoid)** . *Let $S$ be an ellipsoid in $\mathbb{R}^D$ defined by the positive definite matrix $\mathbf{H} \in \mathbb{R}^{D\times D}$:*

$$S(\epsilon) := \{\mathbf{w} \in \mathbb{R}^D : \frac{1}{2}\mathbf{w}^T\mathbf{H}\mathbf{w} \le \epsilon\}$$

*Then $w(S)^2$ or the squared Gaussian width of the ellipsoid satisfies:*

$$w(S)^2 \approx 2\epsilon\mathrm{Tr}(\mathbf{H}^{-1}) = \sum_i r_i^2$$

*where $r_i = \sqrt{2\epsilon/\lambda_i}$ with $\lambda_i$ is $i$-th eigenvalue of $\mathbf{H}$.*

$Proof.$ Let $\mathbf{g} \sim \mathcal{N}(\mathbf{0}, \mathbf{I}_{D \times D})$ and $\mathbf{L}\mathbf{L}^T = 2\epsilon\mathbf{H}^{-1}$. Then:

$$\begin{aligned}
w(L(B_2^n)) &= \frac{1}{2}\mathbb{E}\, sup_{\mathbf{x},\mathbf{y}\in B(0,1)} < \mathbf{g}, \mathbf{L}\mathbf{x} - \mathbf{L}\mathbf{y} > \\
&= \frac{1}{2}\mathbb{E}\, sup_{\mathbf{x},\mathbf{y}\in B(0,1)} < \mathbf{L}^{\mathbf{T}}\mathbf{g}, \mathbf{x} - \mathbf{y} > \\
&= \mathbb{E}\|\mathbf{L}^{\mathbf{T}}\mathbf{g}\|_2 && \text{Definition of Ball.} \\
&= \mathbb{E}\sqrt{\mathbf{g}^{\mathbf{T}}\mathbf{L}\mathbf{L}^{\mathbf{T}}\mathbf{g}} && \|\mathbf{g}\|_2 = \sqrt{\mathbf{g}^{\mathbf{T}}\mathbf{g}}, \text{ where } \mathbf{g} \in \mathbb{R}^{D\times 1}. \\
&= \mathbb{E}\sqrt{2\epsilon\mathbf{g}^{\mathbf{T}}\mathbf{H}^{-1}\mathbf{g}} \\
&\approx \sqrt{2\epsilon\mathbb{E}[\mathbf{g}^{\mathbf{T}}\mathbf{H}^{-1}\mathbf{g}]} && \text{Lemma } D.2. \\
&= \sqrt{2\epsilon\text{Tr}(\mathbf{H}^{-1})} && \text{Invariance of Gaussian under rotation.}
\end{aligned}$$

Thus, $w(S)^2 \approx 2\epsilon\text{Tr}(\mathbf{H}^{-1}) = \sum_i^D r_i^2$.

## D.2 Gaussian Width of the Deformed Ellipsoid

Generally, it is assumed that the gradient descent algorithm will converge to a minimum point. However, in practice, even with small learning rates, the network may oscillate near the minimum point and not directly converge to it, but rather get very close to it. As a result, the actual Hessian matrix is often not positive definite and its eigenvalues may have negative values.

**Lemma D.6** *Let the Hessian matrix at the minimum point be denoted by* $\mathbf{H}$ *with eigenvalue* $\lambda_i$, *and the Hessian matrix at an oscillation point be denoted by* $\hat{\mathbf{H}}$ *with eigenvalue* $\hat{\lambda}_i$. *The negative eigenvalues of* $\hat{\mathbf{H}}$ *have small magnitudes.*

$Proof.$ Let the weights at the minimum point be denoted by $\mathbf{w}_0$ and the Hessian matrix at an oscillation point be denoted by $\hat{\mathbf{w}}_0$. Consider a loss function $\mathcal{L}$ and a loss landscape defined by $\mathcal{L}(\mathbf{w})$, taking Taylor expansion of $\mathcal{L}(\mathbf{w})$ at $\mathbf{w}_0$:

$$\mathcal{L}(\mathbf{w}) = \mathcal{L}(\mathbf{w}_0) + \frac{1}{2}(\mathbf{w} - \mathbf{w}_0)^T\mathbf{H}(\mathbf{w} - \mathbf{w}_0) + R(\mathbf{w}_0)$$

Let $\hat{\mathbf{w}}_\mathbf{0} = \mathbf{w}_0 + \mathbf{v}$ with $\mathbf{v}$ is closed to $\mathbf{0}$:

$$\begin{aligned}
\mathcal{L}(\hat{\mathbf{w}}_\mathbf{0}) &= \mathcal{L}(\mathbf{w}_0 + \mathbf{v}) \\
&= \mathcal{L}(\mathbf{w}_0) + \frac{1}{2}\mathbf{v}^T\mathbf{H}\mathbf{v} + R(\mathbf{w}_0 + \mathbf{v})
\end{aligned}$$

Therefore, the second order derivative of $\mathcal{L}(\hat{\mathbf{w}}_\mathbf{0})$ is:

$$\begin{aligned}
\mathcal{L}''(\mathbf{w}) &= \mathcal{L}''(\mathbf{w}_0 + \mathbf{v}) \\
&= \mathbf{H} + R''(\mathbf{w}_0 + \mathbf{v}) \\
&\approx \mathbf{H}
\end{aligned}$$

where $\mathcal{L}''(\mathbf{w}) = \hat{\mathbf{H}}$, Let $\mathbf{H} = \hat{\mathbf{H}} + \mathbf{H}_0$ with $\mathbf{H}_0$ is closed to $\mathbf{0}$, considering the Weyl inequality:

$$\lambda_i(\mathbf{H}) - \hat{\lambda}_i(\hat{\mathbf{H}}) \leq \|\mathbf{H}_0\|_2$$

where $\|\mathbf{H}_0\|_2$ is small enough. So if $\hat{\lambda}_i(\hat{\mathbf{H}})$ is less than 0, since $\hat{\lambda}_i(\hat{\mathbf{H}}) \geq \lambda_i(\mathbf{H}) - \|\mathbf{H}_0\|_2$, its absolute value $|\hat{\lambda}_i(\hat{\mathbf{H}})| \leq \|\mathbf{H}_0\|_2 - \lambda_i(\mathbf{H}) \leq \|\mathbf{H}_0\|_2$, which means that the negative eigenvalues of the Hessian matrix have small magnitudes.

**Lemma D.7** *For a sublevel set* $S(\epsilon) := \{\mathbf{w} : \mathbf{w}^T\mathbf{H}\mathbf{w} \leq \epsilon\}$ *defined by a matrix* $\mathbf{H}$ *with small magnitude negative eigenvalues. The Gaussian width of* $S(\epsilon)$ *can be estimated by obtaining the absolute values of the eigenvalues of the matrix* $\mathbf{H}$.

$Proof.$ Assuming that the eigenvalue decomposition of $\mathbf{H}$ is $\mathbf{H} = \mathbf{v}^T \boldsymbol{\Sigma} \mathbf{v}$, where $\boldsymbol{\Sigma}$ is a diagonal matrix consisting of the eigenvalues of $\mathbf{H}$, let $\mathbf{D} = \mathbf{v}^T |\boldsymbol{\Sigma}| \mathbf{v}$ be a positive definite matrix and $\mathbf{M} = \mathbf{H} - \mathbf{D} = \mathbf{v}^T (\boldsymbol{\Sigma} - |\boldsymbol{\Sigma}|) \mathbf{v}$ be a negative definite matrix. Consider the definition of $S(\epsilon)$:

$$\begin{aligned}
\mathbf{w}^T \mathbf{H} \mathbf{w} &= \mathbf{w}^T (\mathbf{H} - \mathbf{D} + \mathbf{D}) \mathbf{w} \\
&= \mathbf{w}^T \mathbf{M} \mathbf{w} + \mathbf{w}^T \mathbf{D} \mathbf{w} \\
&\leq \epsilon
\end{aligned}$$

Therefore, $S(\epsilon)$ can be expressed as $\mathbf{w}^T \mathbf{D} \mathbf{w} \leq \epsilon - \mathbf{w}^T \mathbf{M} \mathbf{w}$. Since the magnitudes of the negative eigenvalues of $\mathbf{H}$ are very small, we can assume that $\mathbf{w}^T \mathbf{M} \mathbf{w}$ is also small, and thus $\mathbf{w}^T \mathbf{D} \mathbf{w} \leq \epsilon - \mathbf{w}^T \mathbf{M} \mathbf{w}$ can be approximately equal to $\mathbf{w}^T \mathbf{D} \mathbf{w} \leq \epsilon$. As a result, we can estimate the Gaussian width of $S(\epsilon)$ by approximating it with the absolute values of the eigenvalues of $\mathbf{H}$.

**Corollary D.8** *Consider a well-trained neural network with weights $\mathbf{w}$, whose loss function defined by $\mathbf{w}$ has a Hessian matrix $\mathbf{H}$. Due to the non-positive definiteness of $\mathbf{H}$, there exist negative eigenvalues. Let the eigenvalue decomposition of $\mathbf{H}$ be $\mathbf{H} = \mathbf{v}^T \boldsymbol{\Sigma} \mathbf{v}$, where $\boldsymbol{\Sigma}$ is a diagonal matrix of eigenvalues. Let $\mathbf{D} = \mathbf{v}^T |\boldsymbol{\Sigma}| \mathbf{v}$, where $|\cdot|$ means absolute operation. the geometric objects defined by H and D are $S(\epsilon) := \{\mathbf{w} \in \mathbb{R}^D : \frac{1}{2} \mathbf{w}^T \mathbf{H} \mathbf{w} \leq \epsilon\}$ and $\hat{S}(\epsilon) := \{\mathbf{w} \in \mathbb{R}^D : \frac{1}{2} \mathbf{w}^T \mathbf{D} \mathbf{w} \leq \epsilon\}$, then the gaussian width of the two set satisfy:*

$$w(S(\epsilon)) \approx w(\hat{S}(\epsilon))$$

# E    Comparison between the Upper and Lower Bound

This section provided the proofs used in the upper bound derivation and roughly analyzed how the lower bound changes when the upper bound varies.

## E.1    $D - k$ Dimension Sublevel Set is Still an Ellipsoid

In the derivation of the upper bound for the pruning ratio threshold, we employed a $D - k$ dimensional loss sublevel set:

$$S(\mathbf{w}^{'}) = \{\mathbf{w}^{'} \in \mathbb{R}^{D-k} : \mathcal{L}([\mathbf{w}^1, \mathbf{w}^{'}]) \leq \mathcal{L}(\mathbf{w}^*) + \epsilon\} \tag{23}$$

Perform Taylor expansion to $\mathcal{L}([\mathbf{w}^1, \mathbf{w}^{'}])$ with respect to $\mathbf{w}^{'}$, the sublevel set is represented as:

$$S(\mathbf{w}^{'}) = \{\mathbf{w}^{'} \in \mathbb{R}^{D-k} : \frac{1}{2}(\mathbf{w}^{'})^T \mathbf{H}^{'} \mathbf{w}^{'} \leq \epsilon\} \tag{24}$$

where $\mathbf{H}^{'}$ is the Hessian matrix of $\mathcal{L}([\mathbf{w}^1, \mathbf{w}^{'}])$ with respect to $\mathbf{w}^{'}$.

Given that the full sublevel set $S(\epsilon) = \{\mathbf{w} \in \mathbb{R}^D : \frac{1}{2}\mathbf{w}^T \mathbf{H} \mathbf{w} \leq \epsilon\}$ is an ellipsoid body, which implies that $\mathbf{H}$ is a positive definite matrix, it is evident that $\mathbf{H}^{'}$ is the principal submatrix of $\mathbf{H}$. Consequently, $\mathbf{H}^{'}$ is also a positive definite matrix, which implies that the sublevel set $S(\mathbf{w}^{'})$ remains an ellipsoid.

## E.2    Relationship between the Upper and Lower Bound

**Theorem E.1 (Eigenvalue Interlacing Theorem)** *Suppose* $\mathbf{A} \in \mathbb{R}^{n \times n}$ *is symmetric, Let* $\mathbf{B} \in \mathbb{R}^{m \times m}$ *with* $m < n$ *be a principal submatrix(obtained by deleting both $i$-th row and $i$-th column for some values of $i$). Suppose* $\mathbf{A}$ *has eigenvalues* $\lambda_1 \leq \cdots \leq \lambda_n$ *and* $\mathbf{B}$ *has eigenvalues* $\beta_1 \leq \cdots \leq \beta_m$. *Then*

$$\lambda_k \leq \beta_k \leq \lambda_{k+n-m} \quad for \quad k = 1, \ldots, m \tag{25}$$

*And if* $m = n - 1$,

$$\lambda_1 \leq \beta_1 \leq \lambda_2 \leq \beta_2 \leq \cdots \leq \beta_{n-1} \leq \lambda_n \tag{26}$$

Next, we provide an elucidation on the relationship between the upper bound and lower bound variations:

**Lemma E.2** *The direct and straightforward relationship between the upper bound and the lower bound can be articulated as follows:*

1. *When the eigenvalues change, the upper and lower bounds will change in the same direction;*

2. *When the weight magnitude changes, the upper bound will change in the same direction as the upper bound or do not change.*

$Proof.$ 1. When the eigenvalues change, the upper and lower bounds will change in the same direction;

By leveraging the Eigenvalue Interlacing Theorem (Theorem E.1), the eigenvalues of the principal submatrix in the upper bound is bounded by the eigenvalues in the lower bound. It's obvious that if the eigenvalues in the lower bound change, the eigenvalues in the upper bound will also change in the same direction, leading to the upper and lower bounds will change in the same direction.

2. When the weight magnitude changes, the upper bound will change in the same direction as the lower bound or do not change.

It's noted that the number of weights in the lower bound is more than the one of weights in the upper bound. These weights are used to calculate the projection distance. So it's clear that when the weight magnitude changes, the upper bound will change in the same direction as the lower bound or not change.

# F   Sharpness & Lower Bound

In this section, we will provide the relationship between the lower bound of the pruning ratio and the sharpness of the loss landscape w.r.t the weights. We first introduce the definition of sharpness, which is similar to the sharpness definition in [26] and [9]:

**Definition F.1** *Given a second-order derivable function $f(\mathbf{w})$, where $\mathbf{w}$ is the parameters. Considering a hessian matrix $\mathbf{H}$ w.r.t. parameters $\mathbf{w}$, the sharpness of $f(\mathbf{w})$ w.r.t. parameters is defined as the trace of $\mathbf{H}$, i.e. $\mathrm{Tr}(\mathbf{H})$.*

As defined in definition F.1, a smaller trace indicates a flatter function. Next, we will provide the connection between the sharpness and the lower bound:

**Lemma F.2** *Given a well-trained neural network $f(\mathbf{w}, \mathbf{x})$, where $\mathbf{w}$ is the parameters and the $\mathbf{x}$ is the input. The lower bound of pruning ratio $\rho_l$ and the sharpness $\mathrm{Tr}(\mathbf{H})$ obeys:*

$$\rho_l \leq 1 - \frac{2\epsilon D}{\|\mathbf{w}^* - \mathbf{w}^k\|_2^2 \mathrm{Tr}(\mathbf{H}) + 2\epsilon D} \tag{27}$$

*where $\mathbf{H}$ is the hessian matrix of $f(\mathbf{w})$ w.r.t. $\mathbf{w}$.*

$Proof.$

$$
\begin{aligned}
\rho_l &= 1 - \frac{1}{D} \sum_{i=1}^{D} \frac{r_i^2}{\|\mathbf{w}^* - \mathbf{w}^k\|_2^2 + r_i^2} \\
&= 1 - \frac{2\epsilon}{D} \sum_{i=1}^{D} \frac{1}{\|\mathbf{w}^* - \mathbf{w}^k\|_2^2 \lambda_i + 2\epsilon} \\
&\leq 1 - \frac{2\epsilon}{D} \frac{D^2}{\sum_{i=1}^{D}(\|\mathbf{w}^* - \mathbf{w}^k\|_2^2 \lambda_i + 2\epsilon)} \qquad \text{Cauchy–Schwarz Inquality.} \\
&= 1 - \frac{2\epsilon D}{\|\mathbf{w}^* - \mathbf{w}^k\|_2^2 \sum_{i=1}^{D} \lambda_i + 2\epsilon D} \\
&= 1 - \frac{2\epsilon D}{\|\mathbf{w}^* - \mathbf{w}^k\|_2^2 \mathrm{Tr}(\mathbf{H}) + 2\epsilon D}
\end{aligned}
$$

It's obvious that if the trace of the hessian matrix becomes smaller, the lower bound will also decrease, indicating a higher sparsity level. Utilizing sharpness as the optimization objective for network pruning is both a rational and efficacious approach.

**Corollary F.3** *Given a well-trained neural network $f(\mathbf{w}, \mathbf{x})$, where $\mathbf{w}$ is the parameters and the $\mathbf{x}$ is the input. The pruning ratio lower bound and the sharpness obeys:*

$$\rho_l \leq 1 - \frac{2\epsilon D}{\|\mathbf{w}^* - \mathbf{w}^k\|_2^2 \mathrm{Tr}(\mathbf{H}) + 2\epsilon D} \tag{28}$$

*where $\mathbf{H}$ is the hessian matrix of $f(\mathbf{w})$ w.r.t. $\mathbf{w}$. An informal version of this corollary can be stated as: sharpness controls the lower bound of the pruning ratio, specifically, a flatter neural network can be pruned more sparsely.*

# G Full Results

Here we present the full set of experiments performed for the results in the main text.

## G.1 The Distance Between the Distribution of Weights

In this section, we provide the total variation(TV) distance between the distribution of trained weights with independent initialization.

Table 9: The TV Distance Between the Distribution of Weights.

| CIFAR10 | FC5 | FC12 | Alexnet | VGG16 |
|---|---|---|---|---|
| TV | 0.02±0.01 | 0.01±0.001 | 0.02±0.02 | 0.01±0.008 |

| ResNet | 18 on CIFAR100 | 50 on CIFAR100 | 18 on TinyImagenet | 50 on TinyImagenet |
|---|---|---|---|---|
| TV | 0.04±0.04 | 0.03±0.02 | 0.02±0.03 | 0.03±0.02 |

## G.2 Comparison of Pruning Algorithms

As discussed in Section 4, we adopt $l_1$ regularization during the training phase, and the hyper-parameter for the $l_1$ regularization is selected empirically. After thorough training, we applied magnitude pruning to reduce the network to the lowest pruning ratio at which the network's performance is maintained, and we didn't apply fine-tuning after pruning.

We validated $l_1$ regularized one-shot magnitude pruning algorithm(LOMP) against four baselines: dense weight training and three pruning algorithms: (i) Rare Gems(RG) proposed by [28], (ii) Iterative Magnitude Pruning(IMP) donated by [7], (iii) Smart-Ratio (SR) which is the random pruning method proposed by [29]. Table 10 shows the pruning performance of the above algorithms, our pruning algorithm is better performing than other algorithms.

Table 10: Performance comparison of various pruning algorithms.

| Dataset | Model | Dense Acc (%) | Sparsity (%) | Test Acc (%)@top-1 | | | |
|---|---|---|---|---|---|---|---|
| | | | | LOMP(ours) | RG | IMP | SR |
| | FC5 | 55.3±0.62 | 1.7 | **59.96±0.45** | 58.76±0.15 | 58.83±0.24 | - |
| CIFAR10 | FC12 | 55.5±0.26 | 1.0 | **60.84±0.21** | 54.96±0.28 | 59.37±0.21 | - |
| | VGG16 | 90.73±0.22 | 0.6 | **91.66±0.08** | 88.76±0.13 | 89.22±0.24 | 87.32±0.11 |
| CIFAR100 | ResNet18 | 72.19±0.23 | 1.9 | **71.82±0.09** | 69.33±0.22 | 68.55±0.21 | 65.74±0.27 |
| | ResNet50 | 74.07±0.43 | 2.0 | **75.22±0.11** | 72.21±0.25 | 69.02±0.23 | 68.58±0.33 |
| TinyImageNet | ResNet18 | 52.92±0.13 | 4.2 | **55.42±0.02** | 45.43±0.27 | 45.12±0.19 | 43.28±0.21 |
| | ResNet50 | 56.45±0.17 | 2.9 | **57.49±0.01** | 51.41±0.28 | 46.93±0.41 | 40.42±0.31 |

**Remark.** To demonstrate the effectiveness of $l_1$ regularization, the dense performance is obtained by training without any regularization. we obtained the dense performance by training without any regularization. According to our theoretical findings, LOMP is the optimal pruning strategy in the pruning-after-training regime, as the $l_1$ is the closest convex relaxation to the $l_0$ regularization, and the magnitude pruning can achieve the lowset pruning ratio. Consequently, we compared LOMP with several pruning algorithms that are not part of the pruning-after-training regime. Notably, LOMP outperforms these other pruning algorithms.

**Discussion.** The $l_1$ regularization metric is not new, and in practice, it becomes increasingly challenging to empirically select the hyper-parameter for $l_1$ regularization and train models with $l_1$ regularization as the model size grows. Nevertheless, some reparametrization techniques to address the $l_1$ regularization issue, as discussed in [44], may facilitate the use of $l_1$ regularization for pruning, making $l_1$ regularization one-shot pruning a very promising approach.

## G.3 Comparison of Pruning as Optimization

In this section, we compared our $l_1$ regularization-based one-shot magnitude pruning with those works treating pruning as optimization [4; 40; 41]. Our results are obtained by searching the hyper-

parameters of $l_1$ regularization, all the data and model settings are followed from [4]. The comparison results can be found in Table 11. As is shown in Table 11, in extremely high levels of pruning schemes, our proposed method performs better than other methods.

Table 11: The pruning performance (model accuracy) of various methods on MLPNet, ResNet20, and ResNet50. As to the performance of MP, WF, CBS, CHITA, and EWR, we adopt the results reported in [4]. We take five runs for our approaches and report the mean and standard error (in the brackets). The best accuracy values (significant) are highlighted in bold. Here sparsity denotes the fraction of zero weights in convolutional and dense layers.

| Network | Sparsity | MP | WF | CBS | CHITA | EWR | LOMP(ours) |
|---|---|---|---|---|---|---|---|
| MLPNet on MNIST (93.97%) | 0.5 | 93.93 | 94.02 | 93.96 | 95.97 (±0.05) | 95.24 (±0.03) | **97.43(±0.23)** |
| | 0.6 | 93.78 | 93.82 | 93.96 | 95.93 (±0.04) | 95.13 (±0.01) | **97.42(±0.22)** |
| | 0.7 | 93.62 | 93.77 | 93.98 | 95.89 (±0.06) | 95.05 (±0.04) | **97.45(±0.24)** |
| | 0.8 | 92.89 | 93.57 | 93.90 | 95.80 (±0.03) | 94.84 (±0.03) | **97.53(±0.21)** |
| | 0.9 | 90.30 | 91.69 | 93.14 | 95.55 (±0.03) | 94.30 (±0.05) | **97.61(±0.07)** |
| | 0.95 | 83.64 | 85.54 | 88.92 | 94.70 (±0.06) | 92.86 (±0.05) | **96.85(±0.14)** |
| | 0.98 | 32.25 | 38.26 | 55.45 | 90.73 (±0.11) | 85.71 (±0.09) | **92.90(±0.67)** |
| ResNet20 on CIFAR10 (91.36%) | 0.5 | 88.44 | 90.23 | 90.58 | 91.04 (±0.09 | **92.04 (±0.03)** | 91.05 (±0.07) |
| | 0.6 | 85.24 | 87.96 | 88.88 | 90.78 (±0.12) | **91.98 (±0.09)** | 91.05 (±0.06) |
| | 0.7 | 78.79 | 81.05 | 81.84 | 90.38 (±0.10) | **91.89 (±0.10)** | 90.82 (±0.09) |
| | 0.8 | 54.01 | 62.63 | 51.28 | 88.72 (±0.17) | 90.15 (±0.09) | **90.28 (±0.06)** |
| | 0.9 | 11.79 | 11.49 | 13.68 | 79.32 (±1.19) | 88.82 (±0.10) | **89.26 (±0.28)** |
| | 0.95 | - | - | - | - | 81.33 (±0.15) | **86.16 (±0.11)** |
| | 0.98 | - | - | - | - | 69.21 (±0.24) | **79.15 (±0.26)** |
| ResNet50 on CIFAR10 (92.78%) | 0.95 | - | - | - | - | 84.96 (±0.15) | **93.62 (±0.16)** |
| | 0.98 | - | - | - | - | 82.85 (±0.20) | **89.55 (±0.58)** |

### G.4 Small Weights Benefits Pruning

We verify that small sharpness is not equal to high sparsity through hypothetical experiments. Considering that the hessian matrix of network $A$ and network $B_1, B_2, B_3, B_4$ share eigenvalues $\{\lambda_1, \lambda_2, \ldots, \lambda_n\}$, the weight magnitude of network $B_1, B_2, B_3, B_4$ is 2,3,4,5 times that of network $A$, we take the eigenvalues and weights from a FC network trained without regularization. In this way, the gap between the curves will be more obvious. For other networks, the trend of the curve gap is consistent, the prediction of the network pruning ratio is shown in the Figure. 7. It is observed from Figure. 7 that as the magnitude of network weights increases, the capacity of the network to tolerate pruning decreases. The pruning ratio threshold is affected not only by loss sharpness but also the magnitude of weights. This finding, on the other hand, provides further evidence of the effectiveness of the $l_1$-norm in pruning tasks.

### G.5 Statistical Information of Weights in Various Experiments

The same plots as Fig. 2(b) and Fig. 2(c) are provided in Figure 8

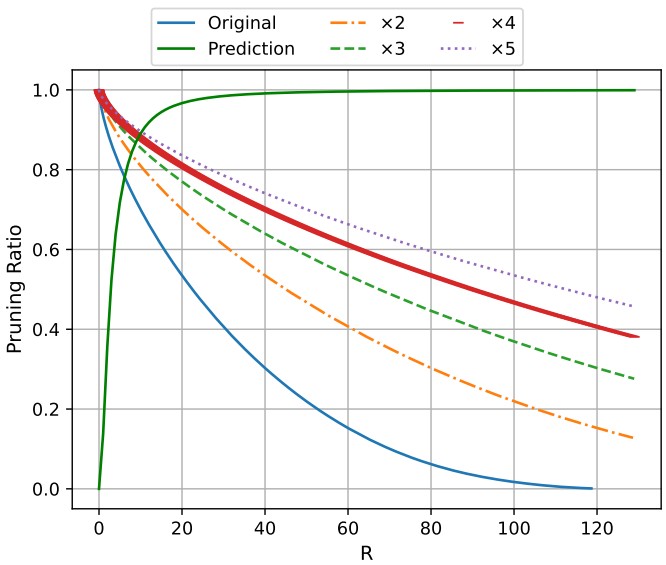

Figure 7: Pruning ratio prediction on different weight magnitude.

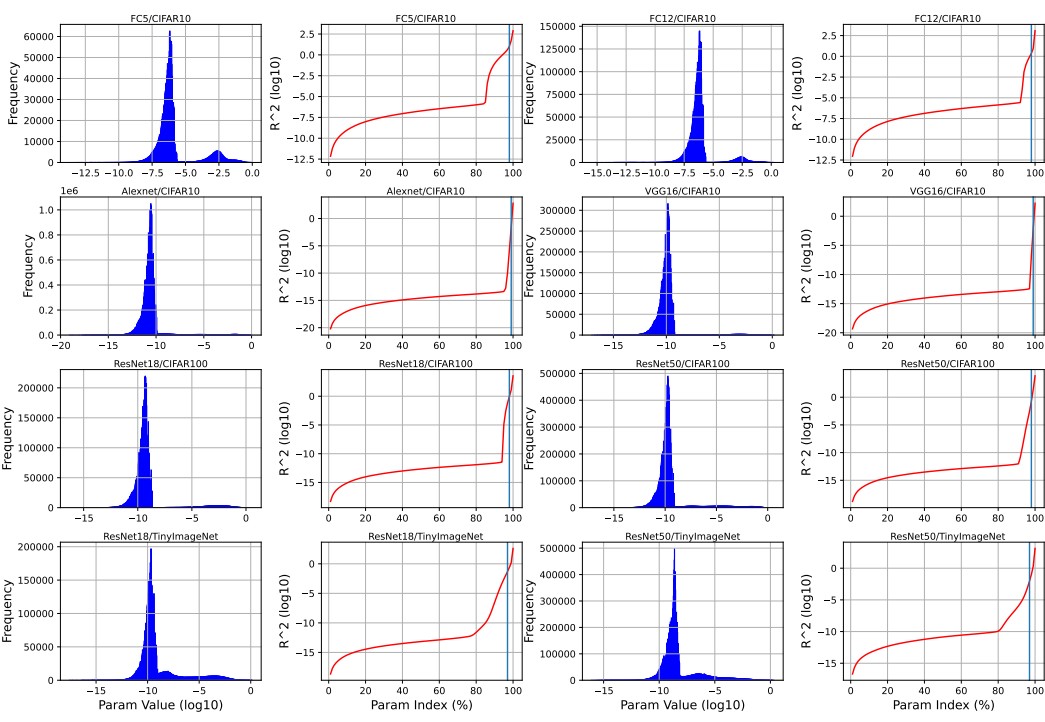

Figure 8: The same plots as Fig. 2(b) and Fig. 2(c) on eight tasks.

## H   Limitations

In this paper, we consider a well-trained neural network and argue that the weight magnitude and network sharpness with respect to the weights constitute the fundamental limits of a one-shot network pruning task. Although popular methods such as Iterative Magnitude Pruning [7] and the Lottery Ticket Hypothesis [6] involve multiple rounds of one-shot pruning, the fundamental limits of such multi-shot pruning remain unclear. Furthermore, we demonstrate that when the magnitude of the removed weights is small, the upper bound of the pruning ratio nearly coincides with the lower bound. However, not all training strategies result in small removed weights, and in such cases, the upper and lower bounds do not coincide. Nevertheless, weight magnitude and network sharpness with respect to the weights still represent the fundamental limits of a one-shot network pruning task.

## I   Broader Impacts

Our work aims to advance the theoretical understanding of network pruning, with the anticipation that theoretical insights can guide future designs of network pruning methods. There are no ethically related issues or negative societal consequences in our work.

