# OpenReview forum: "How Sparse Can We Prune A Deep Network: A Fundamental Limit Perspective"
_NeurIPS.cc/2024/Conference — NeurIPS 2024 poster_

### Official Review · Reviewer_LoL3 · 2024-07-11

**Soundness:** 3
**Presentation:** 3
**Contribution:** 3
**Rating:** 6
**Confidence:** 3

**Summary:**

The authors build on the work of Larsen et. al. [1]  to estimate the fundamental limit of pruning i.e the smallest achievable density of a network by pruning. This is done by estimating the statistical dimension of the neural network and leveraging convex geometry.
Similar to Larsen et. al. [1] the authors provide a lower bound on the smallest density i.e. the sparsest parameter set that still generalizes well. They also provide an upper bound of this density, showing that such a dense network exists.
The authors provide computational methods to estimate these bounds reasonably for a neural network and empirically validate them.
Overall, the work provides a theoretical characterization of the pruning limit of a network.

**Strengths:**

1. The proposed bounds validate that flatter networks allow more parameters to be pruned as well as the use of magnitude based parameter pruning which has consistently outperformed other pruning criteria.
2. Algorithms to compute these bounds are also provided and empirically validated.
3. Experiments across image classifications tasks are provided which align with the computed limits on pruning.

**Weaknesses:**

1. The analysis of Figure 4 is unclear to me. I cannot follow the interpretation of the provided plots to infer that iterative pruning can be beneficial. The provided plots suggest that removing a large fraction of parameters i.e. having a small pruning ratio, increases the value of R, which is not desirable according to the theory, for a dense network.
However, does this also hold for a partially sparse network which is iteratively pruned like in the case of IMP, how does pruning iteratively mitigate this trend?
2. The authors have also not confirmed this insight via experiments with L1 regularization if iterative pruning helps over one-shot pruning. I believe the authors need to give a detailed explanation of Section 7.
3. In addition, can the authors comment on the connection to the pruning fractions shown in Figure 4 to the observations made by Paul et. al. [2] for IMP (Figure 5 in Paul et. al. makes a similar analysis). Is there an optimal pruning fraction for an iterative method like IMP?

The use of the terms pruning ratio, pruning threshold and sparsity are interchangeably used without concretely defining them, which makes the paper difficult to follow. I would urge the authors to correct this and clarify that the pruning ratio implies the number of nonzero parameters left in the network after pruning (it seems so from the derivation?).
The use of sparsity is incorrect in Table 9, sparsity denotes the number of zero parameters in the network.

**Questions:**

See above.

**Limitations:**

This paper provides a theoretical framework to estimate the pruning limit of a neural network, building on recent work in this direction by Paul et. al. [2] and Larsen et. al. [1] via a convex geometry viewpoint, which can potentially be a useful contribution to the sparsity community.
I am happy to increase my score further if my concerns are sufficiently addressed.


[1] Larsen, Brett W., et al. "How many degrees of freedom do we need to train deep networks: a loss landscape perspective." International Conference on Learning Representations. 2021.

[2] Paul, Mansheej, et al. "Unmasking the Lottery Ticket Hypothesis: What's Encoded in a Winning Ticket's Mask?." The Eleventh International Conference on Learning Representations. 2022.

---

> ### Author Rebuttal · Authors · 2024-08-07
>
> Thank you for your careful reading and constructive comments.
>
> **Weakness 1: The analysis of Figure 4 is unclear to me. I cannot follow the interpretation of the provided plots to infer that iterative pruning can be beneficial. The provided plots suggest that removing a large fraction of parameters i.e. having a small pruning ratio, increases the value of R, which is not desirable according to the theory, for a dense network. However, does this also hold for a partially sparse network which is iteratively pruned like in the case of IMP, how does pruning iteratively mitigate this trend?**
>
> In Fig. 4,  in fact, we do not assert that iterative pruning itself is inherently beneficial. Instead, our aim is to demonstrate that in Iterative Magnitude Pruning (IMP), the lower bound of the pruning ratio increases after each pruning iteration. When the previously set pruning ratio is less than the theoretical limit, the pruning ratio should be adjusted to avoid incorrect pruning. Consequently, fewer weights need to be pruned in subsequent iterations, leading to an adjustment in the pruning ratio. This forms the basis for the pruning ratio adjustment algorithm.
>
>
>
> **Weakness 2: The authors have also not confirmed this insight via experiments with L1 regularization if iterative pruning helps over one-shot pruning. I believe the authors need to give a detailed explanation of Section 7.**
>
> We did not compare the performance of iterative pruning with one-shot pruning directly. Instead, our objective is to use our theoretical results to elucidate why iterative pruning requires adjustment of the pruning ratio. Additionally, we aim to explain why, in some iterative pruning algorithms, the use of regularization versus no regularization can lead to discrepancies in the final performance of the sparse network.
>
>
> **Weakness 3: In addition, can the authors comment on the connection to the pruning fractions shown in Figure 4 to the observations made by Paul et. al. [2] for IMP (Figure 5 in Paul et. al. makes a similar analysis). Is there an optimal pruning fraction for an iterative method like IMP?**
>
> Figure 4 in our paper and Figure 5 in Paul et al. utilize the same theoretical results to illustrate that in Iterative Magnitude Pruning (IMP), the network cannot be pruned to the target sparsity directly, necessitating iterative pruning. In contrast, our paper shows that after each pruning step, the lower bound of the pruning ratio increases, indicating that the pruning proportion must be adjusted in iterative pruning. The determination of the optimal pruning ratio for iterative methods is an intriguing research question, and we are actively exploring this area.
>
>
>
> **Weankness 4: The use of the terms pruning ratio, pruning threshold and sparsity are interchangeably used without concretely defining them, which makes the paper difficult to follow. I would urge the authors to correct this and clarify that the pruning ratio implies the number of nonzero parameters left in the network after pruning (it seems so from the derivation?). The use of sparsity is incorrect in Table 9, sparsity denotes the number of zero parameters in the network.**
>
> Thanks for your suggestion, pruning ratio and sparsity imply the number of nonzero parameters left in the network after pruning and we will correct those in the revised version.

---

> > ### Comment · Reviewer_LoL3 · 2024-08-09
> > **Response to rebuttal**
> >
> > Thank you for providing clarifications on W1 and W3, these were especially confusing to me. So if I understand correctly, the number of parameters that can be pruned away reduces in each iteration and this motivates the need for iterative pruning?

---

> > > ### Author Response · Authors · 2024-08-11
> > >
> > > Thank you for your response! Your understanding is correct in that the percentage of removable parameters indeed decreases with the iteration. However, this is  not the motivation for iterative pruning, rather, it just motivates the need of *adaptive* pruning ratio *if iterative pruning is used*. As a consequence, this illustrates the sub-optimality of fixed pruning ratio in IMP.
> > >
> > > As a comparison,  Figure 5 in the work of Paul et al. which you mentioned previously serves to demonstrate the need of iterative pruning if an arbitrary target sparsity (say 2%) is set in the advance and a suboptimal pruning scheme (whose pruning lower bound is 30%, for example) is used, otherwise, the target sparsity cannot be achieved.
> > >
> > >
> > > Furthermore, regarding the fundamental performance comparison between one-shot magnitude pruning and iterative pruning, extensive simulations have  shown that the former can achieve  better performance than the latter, if the regularization coefficient is carefully selected. So it's sensible to conjecture that iterative pruning cannot beat the $l_1$ regularized one-shot magnitude pruning (LOMP) from the perspective of fundamental limit of pruning, i.e., iterative pruning is unnecessary in terms of the fundamental pruning limit. Rigorous proof and demonstrations regarding this conjecture is still under investigation.

---

> > > > ### Comment · Reviewer_LoL3 · 2024-08-12
> > > > **Response to author reply**
> > > >
> > > > Thanks again for the clarification. Now it is clear to me.
> > > > All my concerns have been addressed by the authors and hence, I will keep my score.

---

> > > > > ### Author Response · Authors · 2024-08-14
> > > > >
> > > > > Thank you very much again for your time, support, and constructive comments, which have helped to improve the quality of our paper.

---

### Official Review · Reviewer_byyM · 2024-07-12

**Soundness:** 3
**Presentation:** 2
**Contribution:** 2
**Rating:** 5
**Confidence:** 3

**Summary:**

This paper leverages the framework of statistical dimension in convex geometry to characterize the sharp phase transition point, i.e., the fundamental limit of the pruning ratio. Two key factors are found to be important for pruning, weight magnitude and network flatness. The flatter the loss landscape or the smaller the weight magnitude, the smaller pruning ratio. The theoretical analysis also aligns well with the empirical results, demonstrating the effectiveness of the theory proof.

**Strengths:**

1. The powerful approximate kinematics formula in convex geometry is leveraged to provide a very tight characterization of the fundamental limit of the network pruning.

2. It is quite impressive to see the theoretical results perfectly coincide with the experiments.

3. Moreover, the theoretical study can be explored to two common tricks in pruning, gradually pruning and $l_2$ regularization in Rare Gems.

4. Code is provided.

**Weaknesses:**

1. One thing that sounds strange to me is that one conclusion is that "The smaller the network flatness (defined as the trace of the Hessian matrix), the more we can prune the network", which is contrary to the previous common belief, i.e., the flatter the network's landscape, the easier to be pruned. https://arxiv.org/pdf/2205.12694. Could the authors clarify these two counterarguments?

2. While the theoretical proof looks sound, the empirical results in this paper look strange to me.  For instance, it is weird to see that the accuracy maintains the same at a 100% pruning ratio, but drops significantly as it goes below 2%. Does the pruning ratio here stand for a different thing than the common definition, i.e., the ratio of weights that are zero?

3. Can the authors explain why RN50 achieves worse accuracy than RN18 on TinyImageNet?

4. Which pruning algorithm is used for Figure 3? It is surprising to see that we can preserve the original performance at 2% ratio, for RN50 on TinyImageNet.

**Questions:**

Please see the above weaknesses.

**Limitations:**

Provided.

---

> ### Author Rebuttal · Authors · 2024-08-07
>
> **Weakness 1: One thing that sounds strange to me is that one conclusion is that "The smaller the network flatness (defined as the trace of the Hessian matrix), the more we can prune the network", which is contrary to the previous common belief, i.e., the flatter the network's landscape, the easier to be pruned. https://arxiv.org/pdf/2205.12694. Could the authors clarify these two counterarguments?**
>
> Thank you for the insightful question. We should indeed use sharpness (defined as “trace” of the Hessian matrix), where smaller sharpness corresponds to greater pruning. This definition will be adopted in the revised version to avoid confusion.
>
> **Weakness 2: While the theoretical proof looks sound, the empirical results in this paper look strange to me. For instance, it is weird to see that the accuracy maintains the same at a 100% pruning ratio, but drops significantly as it goes below 2%. Does the pruning ratio here stand for a different thing than the common definition, i.e., the ratio of weights that are zero?**
>
> We define pruning ratio as the proportion of remaining weights, 100% means no pruning is performed.
>
> **Weakness 3: Can the authors explain why RN50 achieves worse accuracy than RN18 on TinyImageNet?**
>
> I am not sure which data the accuracy refers to in this context. If it refers to the test accuracy of the network, then Table 9 shows that the accuracy of RN50 is higher.
>
> **Weakness 4: Which pruning algorithm is used for Figure 3? It is surprising to see that we can preserve the original performance at 2% ratio, for RN50 on TinyImageNet.**
>
> We first train the RN50 model with L1 regularization on TinyImageNet. Following the training, we prune the network by removing the smaller weights. Details of the hyperparameters used during training are provided in the appendix. Additionally, a comparison of the pruning algorithms is included in the appendix. Since L1 regularization-based magnitude pruning is not new technique and not the primary focus of this paper, we have placed it in the appendix.

---

> ### Comment · Reviewer_byyM · 2024-08-13
> **After reading rebuttal**
>
> I thank the authors for their response. Defining the pruning ratio as the proportion of remaining weights makes no sense to me. I suggest to change it to the opposite. I am not an expert in the theoretical study of pruning/sparsity. I will keep my original score and defer the assessment of the theoretical analysis to the AC and other reviewers.

---

> > ### Author Response · Authors · 2024-08-14
> >
> > Thank you very much for your time and constructive comments. We would like to clarify that it is acceptable to use either the proportion of remaining weights or the proportion of removed weights as the pruning ratio. When the latter is used, the theoretical result will be 1 minus our theoretical result, which does not affect the correctness of the theory. We will include this clarification in the revised version. Once again, thank you for your time and comments, which have contributed to the improvement of our paper’s quality.

---

### Official Review · Reviewer_hJmV · 2024-07-12

**Soundness:** 3
**Presentation:** 2
**Contribution:** 2
**Rating:** 6
**Confidence:** 3

**Summary:**

The paper investigates the theoretical limits of how sparse a network can be pruned without sacrificing performance. The authors formulate pruning as a convex geometry problem. By imposing sparsity constraints on the loss function, they show that the pruning ratio can be bounded by the width of the loss landscape and the magnitude of the weights. They also propose an spectrum estimation algorithm for dealing with large Hessian matrices. The empirical results align well with the theoretical findings.

**Strengths:**

* Understanding the fundamental limits of network pruning is an important problem, and the insights provided by this paper (weight magnitude and network flatness as key factors) could potentially influence future research in the area.
* The authors have provided detailed proofs and derivations for the theoretical results.
* The experiments conducted are extensive enough to validate the theoretical findings, and there is a strong alignment between the theoretical and empirical results.

**Weaknesses:**

* Limited technical novelty: applying convex geometry to attain similar results (e.g. bounds dependent on the Gaussian width) has been explored in prior works, some of which are cited in the related work section. However, the paper lacks discussion of how this work differs from or builds upon existing literature.
* More intuitions on the theoretical results would be helpful. For example, loss flatness is one of the key factors discussed in the paper, and only its formal definition (trace of the Hessian) is given in the paper. Intuitions like smaller flatness (i.e. flatter loss landscape) means the loss function is less sensitive to perturbations in the weights would be helpful.
* Inconsistencies in notations: For example, from equation (17) onwards, the loss sublevel set notation takes in the weights as the argument. Is this the same loss sublevel set defined in equation (2), which takes in epsilon as the argument? And how is the epsilon parameter chosen for the loss sublevel set?
* I would define "pruning ratio" earlier in the paper. It was not clear until the contributions section that smaller pruning ratios means more pruning of the network.
* The authors have introduced "pruned ratio", which equals to 1 - pruning ratio, for Table 2. This is the only place in the paper where this term is used. This is confusing. Why did the authors not stick to "pruning ratio" throughout the paper?
* Please consider removing adjectives like "powerful" when describing convex geometry and approximate kinematics formula, or "very tight characterization." They are subjective and do not add much to the clarity of the paper.
* Some typos like "Lottery Tickets Hypolothsis" (line 77)
* Some terms like SLQ (line 82) are not defined until later in the paper.

**Questions:**

* What is novel here in light of prior work showing results connecting Gaussian widths to recovery thresholds [4]?
* How sensitive are the theoretical bounds to the choice of loss function? Do the results generalize across different loss functions commonly used in deep learning?
* How computationally expensive is the proposed spectrum estimation algorithm compared to existing methods? Is it practical for very large networks?

**Limitations:**

Yes.

---

> ### Author Rebuttal · Authors · 2024-08-07
>
> Thanks for your constructive feedback! The concerns and questions in the review are addressed as follows.
>
> **Weakness 1: Limited technical novelty: applying convex geometry to attain similar results (e.g. bounds dependent on the Gaussian width) has been explored in prior works, some of which are cited in the related work section. However, the paper lacks discussion of how this work differs from or builds upon existing literature.**
>
> Although previous work has employed convex geometry tools to derive lower bounds for network pruning, these bounds might be far from the actual pruning limit. In contrast, we employ more powerful tools in convex geometry, i.e., the statistical dimension and approximate kinematic formula, thus enabling us to obtain a sharp threshold of pruning.
> Moreover, our research tackles the pruning problem in a more systematic way, and through our theoretical results, we explicitly identify two key factors that determine the pruning limit, i.e., the network flatness and the weight magnitude, which is, in our opinion, valuable to better understand the existing pruning algorithms and design new and more efficient algorithms.
>
>
>
> **Weakness 2: More intuitions on the theoretical results would be helpful. For example, loss flatness is one of the key factors discussed in the paper, and only its formal definition (trace of the Hessian) is given in the paper. Intuitions like smaller flatness (i.e. flatter loss landscape) means the loss function is less sensitive to perturbations in the weights would be helpful.**
>
> Thank you for the suggestion. In the revised version, we will include more intuitive explanations to enhance the understanding of the paper.
>
>
>
> **Weakness 3: Inconsistencies in notations: For example, from equation (17) onwards, the loss sublevel set notation takes in the weights as the argument. Is this the same loss sublevel set defined in equation (2), which takes in epsilon as the argument? And how is the epsilon parameter chosen for the loss sublevel set?**
>
> These are, in fact, two sublevel sets. Consequently, epsilon should be included in the sublevel set notation of Equation 17; we will correct this in the revised version. We compute the standard deviation of the loss across different batches of the training set and use it as epsilon.
>
>
>
> **Weakness 4: I would define "pruning ratio" earlier in the paper. It was not clear until the contributions section that smaller pruning ratios means more pruning of the network.**
>
> Thanks for your suggestion, we will define pruning ratio earlier in the revised version.
>
>
>
> **Weakness 5: The authors have introduced "pruned ratio", which equals to 1 - pruning ratio, for Table 2. This is the only place in the paper where this term is used. This is confusing. Why did the authors not stick to "pruning ratio" throughout the paper?**
>
> Good suggestion, we will change it to "pruning ratio" in the revised version.
>
>
>
> **Weakness 6: Please consider removing adjectives like "powerful" when describing convex geometry and approximate kinematics formula, or "very tight characterization." They are subjective and do not add much to the clarity of the paper.**
>
> We will correct this in the revised version.
>
>
>
> **Weakness 7: Some typos like "Lottery Tickets Hypolothsis" (line 77)**
>
> Thanks for pointing out this, we will correct typos in the revised version.
>
>
>
> **Weakness 8: Some terms like SLQ (line 82) are not defined until later in the paper.**
>
> We will define SLQ before using in the revised version.
>
> **Question 1: What is novel here in light of prior work showing results connecting Gaussian widths to recovery thresholds [4]?**
>
> Ref.[4] and our work share the similarity that they take advantage of convex geometry to characterize the threshold of given inference or learning problems. The main difference between them lies in:
>
> 1) Our work extends significantly the application of convex geometry from a specific inference problem to a general learning problem: Ref. [4] addresses in specific the recovery threshold of the *linear* inverse problem, which is a classical statistical *inference* problem. On the other hand, our work tackles a statistical *learning* problem which is of general DNN architecture and *arbitrary loss function*. The methods and results in Ref. [4] just cannot be simply translated to the  problem we address in our work.
>
> 2) The threshold in our work is normally *sharper* than Ref. [4]: Ref. [4] utilizes the Gaussian width to characterize the recovery threshold, which unfortunately can only provide the lower bound (*necessary condition*) since the key result Ref. [4] relies on, i.e. Gordon's escape through a mesh theorem only provide the necessary condition.  In contrast, our work leverage the statistical dimension framework and the accompanying approximate kinematic formula, whose necessary and sufficient condition almost coincides, thus capable of providing sharp thresholds.
>
>
> **Question 2: How sensitive are the theoretical bounds to the choice of loss function? Do the results generalize across different loss functions commonly used in deep learning?**
>
> Our theory only concerns the pruning limits of well-trained models and does not address the loss function during the training process. In other words, our theoretical results apply to one-step pruning for all well-trained models.
>
>
>
> **Question 3: How computationally expensive is the proposed spectrum estimation algorithm compared to existing methods? Is it practical for very large networks?**
>
> In fact, our proposed improved algorithm introduces only a modest increase in computational complexity compared to existing methods. For instance, with a fixed sample size of 100, it requires more computation of only 200 derivatives.  These computations do not necessitate the use of the entire dataset and can be performed on a subset. Furthermore, for very large networks, the improved algorithm provides a more accurate estimation of the eigen spectrum.

---

> > ### Comment · Reviewer_hJmV · 2024-08-12
> >
> > Thank you for your detailed rebuttal addressing the concerns and questions raised. The explanation of the differences compared to [4] is appreciated. The clarifications on how the statistical dimension framework enables deriving sharper bounds are helpful for understanding the paper's contributions. I am slightly raising my score.

---

> > > ### Author Response · Authors · 2024-08-14
> > >
> > > Thank you once again for your time, support, and constructive feedback, as well as the score increase. We sincerely appreciate your input, which has significantly contributed to improving the quality of our paper. We will add clarifications on how the statistical dimension framework enables deriving sharper bounds in our revised version.

---

### Official Review · Reviewer_DBJe · 2024-07-13

**Soundness:** 3
**Presentation:** 2
**Contribution:** 3
**Rating:** 4
**Confidence:** 3

**Summary:**

This paper tries to answer the question of how sparse can a deep neural network be pruned without increasing the loss function. The authors employ high-dimensional geometry tools such as statistical dimension, Gaussian width, and the Approximate Kinematic Formula to derive the lower bound and upper bound of the ratios of weights that can be pruned. The authors also provide an improved algorithm for estimating the eigenvalues of large Hessian matrices, which is used to estimate the Gaussian width.

**Strengths:**

1. This paper provides an interesting perspective for analyzing the limits of network pruning.

2. The tools from convex geometry are novelly applied to study the problem. The derivations of the lower bound and upper bound are solid.

3. The results provide insights into the problem of pruning neural networks and explain certain behaviors from previous work.

4. The experiments show a certain matching between the theoretical bounds and the actual performance.

**Weaknesses:**

1. The derived upper bound and lower bound on the pruning ratio depend on the new weights of the pruned networks. This is strange to me, meaning that the limit of the pruning ratio varies when a different weight is used. This also limits the application of the results.

2. The numerical experiments are performed using the L1 regularized loss function. However, this is not considered in the theoretical results. I wonder if the bounds will be different if the loss function is changed.

3. The one-shot magnitude pruning ignores the interaction between the weights, which has been shown to be suboptimal in some recent work. However, the authors claim that it is optimal for the lower and upper bounds. I believe this is because the L1 regularization is used as the loss function, which already takes that into account. This should be clarified.

4. The authors claim that the upper and lower bounds match each other when the proposed L1 regularized loss function is used for training. This might not hold when the original loss is without L1 regularization. This should also be discussed and not overclaimed in general cases.

5. The numerical comparison did not consider many state-of-the-art pruning methods (which are also pruning after training) such as
[1] R. Benbaki et. al., Fast as CHITA: Neural Network Pruning with Combinatorial Optimization
[2] L. You et. al., SWAP: Sparse Entropic Wasserstein Regression for Robust Network Pruning
[3] X. Yu et. al., The combinatorial brain surgeon: Pruning weights that cancel one another in neural networks

6. Relating to the previous comment, the magnitude pruning method used here minimizes the bounds proposed here, which then is used as a support for their usefulness. However, in many cases, the network is not pruned to its limit.

7. The title is overclaimed. While the bounds proposed here are sometimes useful, they are not fundamental limits since there are approximations and relaxation used along the derivations. Moreover, the upper and lower bounds do not match each other in general.

**Questions:**

Please see the weakness part.

**Limitations:**

Yes, the authors did address the limitations.

---

> ### Author Rebuttal · Authors · 2024-08-07
>
> We sincerely thank you for your insightful feedback. Below are our detailed responses to your concerns:
>
> **Weakness 1: The derived upper bound and lower bound on the pruning ratio depend on the new weights of the pruned networks. This is strange to me, meaning that the limit of the pruning ratio varies when a different weight is used. This also limits the application of the results.**
>
> Good question. In fact, this is the key point in pruning bounds. We aim for the performance of the sparse network to closely match that of the dense network. In other words, the performance of the sparse network is bounded by that of the dense network. If the weights of the dense model change, the performance bound of the sparse network also changes, leading to a different pruning ratio bound.
>
>
>
> **Weakness 2: The numerical experiments are performed using the L1 regularized loss function. However, this is not considered in the theoretical results. I wonder if the bounds will be different if the loss function is changed.**
>
> In fact, we are considering the pruning limit of a well-trained model without special focus on the loss function. Our theoretical results actually apply to all well-trained models, regardless of the loss function.
>
>
>
> **Weakness 3: The one-shot magnitude pruning ignores the interaction between the weights, which has been shown to be suboptimal in some recent work. However, the authors claim that it is optimal for the lower and upper bounds. I believe this is because the L1 regularization is used as the loss function, which already takes that into account. This should be clarified.**
>
> Thank you for your reminder, we will add relevant instructions to the paper: In the absence of L1 regularization, although the theoretical lower bound for magnitude pruning is the lowest, the achievable upper bound for magnitude pruning may not be the lowest. In general, if the optimal algorithm is not clear, magnitude pruning can be considered as a viable choice.
>
> **Weakness 4: The authors claim that the upper and lower bounds match each other when the proposed L1 regularized loss function is used for training. This might not hold when the original loss is without L1 regularization. This should also be discussed and not overclaimed in general cases.**
>
> Great reminder. In fact, we only claimed that the upper and lower bounds match when the proposed L1 regularized loss function is used for training. This might not hold in other cases, and we will make this point more clear  in the revised version.
>
>
>
> **Weakness 5: The numerical comparison did not consider many state-of-the-art pruning methods (which are also pruning after training) such as [1] R. Benbaki et. al., Fast as CHITA: Neural Network Pruning with Combinatorial Optimization [2] L. You et. al., SWAP: Sparse Entropic Wasserstein Regression for Robust Network Pruning [3] X. Yu et. al., The combinatorial brain surgeon: Pruning weights that cancel one another in neural networks.**
>
> In fact, since we proved that the lower bound of magnitude pruning is smaller than that of other pruning methods, and that the upper bound of magnitude pruning is more likely to match the lower bound, we focused solely on magnitude pruning. We will include a comparison with other algorithms in our revised version.
>
> **Weakness 6: Relating to the previous comment, the magnitude pruning method used here minimizes the bounds proposed here, which then is used as a support for their usefulness. However, in many cases, the network is not pruned to its limit.**
>
> This is true; in general, the lower bound for magnitude pruning is the lowest among pruning methods. However, not all upper bounds for magnitude pruning match this lower bound. If we consider the lower bound as the limit, many models in practice will not be pruned  to this extent. Nonetheless, our theoretical results highlight two key factors that determine network pruning limit: the flatness of the loss landscape and the magnitude of network weights.
>
>
>
> **Weakness 7: The title is overclaimed. While the bounds proposed here are sometimes useful, they are not fundamental limits since there are approximations and relaxation used along the derivations. Moreover, the upper and lower bounds do not match each other in general.**
>
> Thank you for your comments. Although the upper and lower bounds generally do not coincide exactly, their gap is indeed negligibly small. Moreover, our theoretical results identify two key factors that influence network pruning: the flatness of the loss landscape and the magnitude of network weights. By constraining these factors, such as using L1 regularization to control magnitude, the upper and lower bounds are more likely to match. Additionally, despite employing approximations in our derivations, the results we obtain is actually nearly exact asymptotically, which is supported by that our theoretical results closely match the simulation outcomes.

---

> > ### Author Response · Authors · 2024-08-11
> >
> > **Weakness 1: The derived upper bound and lower bound on the pruning ratio depend on the new weights of the pruned networks. This is strange to me, meaning that the limit of the pruning ratio varies when a different weight is used. This also limits the application of the results.**
> >
> > Our previous response seems not addressing your concern in the right direction. So we'd like to supplement our response as follows:
> > We acknowledge this is indeed a sharp observation. The pruning limit we provide does depend on the trained weights, however, it does not mean that this limit will change with different weights (due to different initialization, for example). In fact, in our pruning limit formula, the pruning limit only depends on the weights in a *global* way. To be specific, it depends on the projection distance, i.e., the *sum* of the square of the removed weights and the *bulk* spectrum of the associated Hessian matrix. Therefore, if the final weights after training obeys a given distribution (for example, a heavy-tailed distribution) regardless of the initialization (which is supported by both  empirical and theoretical studies, such as Mahoney's work), the pruning limit will be the same. Simulation results  validate this in Table 2.
> >
> > [1] Lynn, Christopher W., Caroline M. Holmes, and Stephanie E. Palmer. "Heavy-tailed neuronal connectivity arises from Hebbian self-organization." Nature Physics 20.3 (2024): 484-491.
> >
> > [2] Martin, Charles H., and Michael W. Mahoney. "Implicit self-regularization in deep neural networks: Evidence from random matrix theory and implications for learning." Journal of Machine Learning Research 22.165 (2021): 1-73.
> >
> > [3] Martin, Charles H., Tongsu Peng, and Michael W. Mahoney. "Predicting trends in the quality of state-of-the-art neural networks without access to training or testing data." Nature Communications 12.1 (2021): 4122.
> >
> > **Weakness 3: The one-shot magnitude pruning ignores the interaction between the weights, which has been shown to be suboptimal in some recent work. However, the authors claim that it is optimal for the lower and upper bounds. I believe this is because the L1 regularization is used as the loss function, which already takes that into account. This should be clarified.**
> >
> > We realize that we have not fully addressed your concern. So we'd like to update our response as follows:
> >
> > You're correct. L1 regularization is very important to ensure that the lower bound and upper bound of pruning limit match. Furthermore, L1 regularization is a very natural relaxation for L0 regularization if weight sparsity is encouraged. In this sense, we think our pruning limit resulting from the L1 regularization is capable of characterizing the fundamental limit of one-shot pruning.

---

> > > ### Comment · Reviewer_DBJe · 2024-08-14
> > > **Response to rebuttal**
> > >
> > > Thank you for the detailed replies! I appreciate the importance of the fundamental limits of pruning a neural network, and this work provides some insights into this problem. However, I still have some doubts about the applicability of these bounds.
> > >
> > > Some comments to your replies:
> > >
> > > Weakness 1. The bounds proposed in this work depend on the sum of the square of the removed weights, this implies that the limit of the pruning depends on the given weights. I don't see how this dependence is in a global way, as different weights will lead to different bounds.  Moreover, the heavy-tailed behavior in the suggested reference refers to the eigenvalue spectrum of the weight matrix, which doesn't support this argument directly.
> > >
> > > Weakness 2. What I was referring to here is that when you train a network with L1 regularization, this does not tell us the pruning limit of the network when trained without L1 regularization. From my understanding, the bounds will be different for different training losses.
> > >
> > > Weaknesses 3 and 4. Yes, I agree that L1 regularization can be used to promote sparsity. But this also comes with the expense of lowering the performance. In some sense, some performance loss in the pruning phase is moved to the training phase, that's why I am questioning the use of L1 regularization as numerical examples. Comparing the performance with and without the regularization is needed to justify its use here.
> > >
> > > Weakness 5. I agree with the argument that magnitude pruning minimizes the bounds, but the numerical results in the references I suggested all show improvements over magnitude pruning.  A comparison with other state-of-the-art pruning methods would be helpful since they are just bounds and there are some approximations along the derivations.
> > >
> > > Weakness 6. The pruning limit depends on $\epsilon$, does that mean in principle one can realize any pruning limit by adjusting $\epsilon$? I wonder if the theory still applies if a small epsilon is chosen.

---

> > > > ### Author Response · Authors · 2024-08-14
> > > >
> > > > Thank you very much for the valuable and insightful feedback. Our responses are as follows:
> > > >
> > > > **Weakness 1: The bounds proposed in this work depend on the sum of the square of the removed weights, this implies that the limit of the pruning depends on the given weights. I don't see how this dependence is in a global way, as different weights will lead to different bounds. Moreover, the heavy-tailed behavior in the suggested reference refers to the eigenvalue spectrum of the weight matrix, which doesn't support this argument directly.**
> > > >
> > > > Regarding the dependence of pruning limit on the weights, it is actually inevitable in the sense that the weights of a DNN  with given architecture are determined by the training data and the task, in other words, for a specific dataset and task, the optimal network (i.e. weights) are fixed. So what really matters is that the optimization procedure, namely it might result in different weights due to different initializations, for example.
> > > >
> > > > What we mean by "the dependence of the pruning limit on the weights in a global way" is in the following sense: the limit only depends on the *sum* of the squares of those smallest weights and on the bulk spectrum of the associated Hessian matrix. Extensive simulations have shown that different initializations will result in nearly the same empirical distribution of the weights and the spectrum of Hessian matrix, therefore leading to the same pruning limit (cf. Table 2). In summary, our pruning limit actually does not depend on the weight initializations, rather, it is an *intrinsic* property of the NN, given the dataset and the task.
> > > >
> > > > Besides, we'd like to thank you for pointing out that the heavy-tailed property of the weight matrix is just about the spectrum, not their entries. So it is not strong enough to defend our claim, rather, we need to resort to the stability of the empirical dist. of the weights and the bulk spectrum of the Hessian matrix, as mentioned above.
> > > >
> > > > **Weakness 2: What I was referring to here is that when you train a network with L1 regularization, this does not tell us the pruning limit of the network when trained without L1 regularization. From my understanding, the bounds will be different for different training losses.**
> > > >
> > > > Yes, we agree that different training loss will lead to different pruning limit, since the different training loss actually corresponding to different networks, after all, DNN are determined by the training data and the training loss.
> > > >
> > > > So the real problem is: whether the training with L1 regularization will change the pruning limit?  We'll elaborate on this point in the next response.
> > > >
> > > > **Weaknesses 3 and 4: Yes, I agree that L1 regularization can be used to promote sparsity. But this also comes with the expense of lowering the performance. In some sense, some performance loss in the pruning phase is moved to the training phase, that's why I am questioning the use of L1 regularization as numerical examples. Comparing the performance with and without the regularization is needed to justify its use here.**
> > > >
> > > > As discussed above, different training loss will normally result in different pruning limit. To explore the fundamental limit of a DNN, we need to employ its original loss, without any modification (say regularization). Unfortunately, this seems nearly impossible due to the intrinsic challenge of ERM, i.e., overfitting. To alleviate overfitting, we must perform some kind of regularization. In this sense, it's hard to explore the pruning limit of the NN in its most original sense. To make progress, we resort to the most direct version of regularization, so as to find the sparsest solution, i.e., the $l_0$ regularization and its convex relaxation $l_1$ regularization. What remains to explore is under what conditions $l_1$ regularization will lead to the same pruning limit as the $l_0$ regularization. This is actually what we're investigating right now.
> > > >
> > > > **Weakness 5: I agree with the argument that magnitude pruning minimizes the bounds, but the numerical results in the references I suggested all show improvements over magnitude pruning. A comparison with other state-of-the-art pruning methods would be helpful since they are just bounds and there are some approximations along the derivations.**
> > > >
> > > > Thank you for pointing out those references. We'll add performance comparisons among the algorithms in those works and our work in the revised version.
> > > >
> > > > **Weakness 6: The pruning limit depends on $\epsilon$, does that mean in principle one can realize any pruning limit by adjusting $\epsilon$? I wonder if the theory still applies if a small epsilon is chosen.**
> > > >
> > > > Our pruning limit monotonically decreases with $\epsilon$, the tolerable loss with pruning, and it actually applies universally, no matter how small $\epsilon$ is. Actually in our simulations, $\epsilon$ takes value of the order $10^{-3}$.

---

> > > > > ### Comment · Reviewer_DBJe · 2024-08-14
> > > > >
> > > > > Thanks again for your detailed replies! I agree with you that finding the limit of network pruning is a hard problem and I do appreciate the theoretical contribution of this work
> > > > > Some further comments:
> > > > >
> > > > > Weakness 1: I am not so sure about the statement that different initializations will result in nearly the same empirical distribution of the weights. Table 2 doesn't seem to support this statement.
> > > > >
> > > > > Weaknesses 2,3 and 4: The bounds provided in the paper should also work for networks trained with the original loss function. To answer whether L1 regularization would change the pruning ratio limit, it would be helpful to compare the limits and the actual pruning performance for both networks trained with and without the L1 regularization.
> > > > >
> > > > > Weakness 6: So when $\epsilon$ decreases further, will the pruning limit eventually lead to a dense network? (e.g. with about 30% of the weights pruned) In that case, the performance of magnitude pruning should be quite far away from other optimization-based methods. I wonder what are the implications of the bounds in those regimes.

---

> ### Author Response · Authors · 2024-08-14
>
> Sincere thanks once again for your feedback. Below is our response:
>
> **Weakness 1: I am not so sure about the statement that different initializations will result in nearly the same empirical distribution of the weights. Table 2 doesn't seem to support this statement.**
>
> The experiments in Table 2 were conducted using independent initializations (with the same initialization method), and the results indicate that their pruning limits are largely consistent. We also performed a statistical analysis of the weight distributions and calculated the total variation distance between them. Below is a summary of the experimental results, which we will include in the revised version. The experiments suggest that independent initializations do not lead to significant differences in the final weight distributions.
>
> For reasons of time, the table is simplified
>
> task | tv distance of weight distribution Mean (standard deviation)
>
> VGG, CIFAR10 | 0.02(0.02)
>
> AlexNet, CIFAR10 | 0.01(0.008)
>
> ResNet18, CIFAR100 | 0.04(0.04)
>
> ResNet50, CIFAR100 | 0.03(0.02)
>
> ResNet18, TinyImageNet | 0.02(0.03)
>
> ResNet50, TinyImageNet | 0.03(0.02)
>
> **Weaknesses 2,3 and 4: The bounds provided in the paper should also work for networks trained with the original loss function. To answer whether L1 regularization would change the pruning ratio limit, it would be helpful to compare the limits and the actual pruning performance for both networks trained with and without the L1 regularization.**
>
> You're right and thank you for your suggestion. The bounds provided in the paper indeed also work for networks trained with the original loss function and we will include experiments related to the original loss in the revised version.
>
> **Weakness 6: So when $\epsilon$ decreases further, will the pruning limit eventually lead to a dense network? (e.g. with about 30% of the weights pruned) In that case, the performance of magnitude pruning should be quite far away from other optimization-based methods. I wonder what are the implications of the bounds in those regimes.**
>
> If $\epsilon$ is significantly reduced, the amount of pruning that can be done will indeed decrease, as our tolerance for performance degradation also becomes smaller. Therefore, $\epsilon$ should not be chosen arbitrarily, as values that are too high or too low are not appropriate. We compute the standard deviation of the loss across different batches of the training set and use it as $\epsilon$.

---

### Author Response · Authors · 2024-08-11
**General Comment to All Reviewers**

We thank all the reviewers for their time and for all of the constructive feedback we have received. We saw some common themes among the reviews, so we'd like to address them before proceeding to respond to the individual reviews.

The key contribution of our work is that we address the network pruning problem from a fundamental limit perspective, i.e., we directly impose the sparsity constraint on the loss function and characterize the corresponding possibly sparsest solution by leveraging the tools from high-dimensional convex geometry, thus enabling a very sharp threshold of network pruning and establishing the fundamental limit of one-shot network pruning for the first time.

The key enabler of the sharp threshold of the network pruning is the notion of statistical dimension and the associated Approximate Kinematic Formula, which provides sufficient and necessary condition, which turns out to be *very tight* in the sense of ratio, i.e., normalization by the ambient dimension.  This is in contrast to the previous works by Ganguli et al. in which the Gaussian width and the Gordon's Escape Theorem are utilized, failing to provide the upper bound due to Gordon's Escape Theorem can only offer the necessary condition (i.e., impossibility result).

From the perspective of application, through our theoretical results we highlight the critical value of the magnitude-based pruning, by demonstrating that it is capable of achieving the smallest possible pruning ratio by rigorous proof. Moreover, we explicitly identify the crucial role that loss flatness (or sharpness) plays in determining the performance of pruning.

---

### Decision · Program_Chairs · 2024-09-25

**Decision:**

Accept (poster)

**Comment:**

This paper studies the prunability of a deep neural network using convex geometry. The main technical tool is the Approximate Kinematic Formula, which characterizes the sharp phase transition between the cases when two convex sets intersect with probability 1 or probability 0, given the two sets' statistical dimensions. This phase transition is then used to obtain the upper and lower bounds of prunability. The bounds using statistical dimensions are converted to a computable bound using Gaussian width. The authors used this bound to find two key contributing factors for prunability, including weight magnitude and network flatness. Then, it was shown that the upper and lower bounds are close to each other when networks are trained with L1 regularization and one-shot magnitude pruning.

There are several reasons that motivate most reviewers (including me) to accept this paper. First, using tools from convex geometry to study prunability is novel. Second, the theories provide insights into important factors to improve network pruning, such as flatness. Finally, the bounds are non-vacuous in specific scenarios, and there is a good alignment between theory and experimental results. While reviewers are concerned with the L1 regularization used in the loss function when the upper and lower bounds are close, I believe this just needs to be stated clearly in the paper. I believe the paper can be improved by stating the assumptions more clearly and perhaps downplaying the "fundamental limit" part slightly. The reviewers also provided other useful suggestions to improve the paper, including comparison with reference [4] and stating the "pruning ratio" clearly.